# Conformal Correction for Efficiency May be at Odds with Entropy

## Abstract

Conformal prediction (CP) provides a comprehensive framework to produce statistically rigorous uncertainty sets for black-box machine learning models. To further improve the efficiency of CP, conformal correction is proposed to fine-tune or wrap the base model with an extra module using a conformal-aware inefficiency loss. In this work, we empirically and theoretically identify a trade-off between the CP efficiency and the entropy of model prediction. We then propose an entropy-constrained conformal correction method, exploring a better Pareto optimum between efficiency and entropy. Extensive experimental results on both computer vision and graph datasets demonstrate the efficacy of the proposed method. For instance, it can significantly improve the efficiency of state-of-the-art CP methods by up to 34.4%, given an entropy threshold.

## 1 Introduction

For a decision-making process driven by machine learning (e.g., loan approval, fraud detection), it is essential for the predictions to be accompanied by a level of confidence to quantify uncertainty (Vovk et al., 2005; Smith, 2024). Conformal prediction (CP) is a promising uncertainty quantification method, providing statistically rigorous uncertainty sets for black-box machine learning models (Babbar et al., 2022; Straitouri et al., 2023; Straitouri & Rodriguez, 2023; Cresswell et al., 2024). In standard classification, for any test input $x$, the posterior distribution $\tilde{\pi}_y(x) = P(Y = y \mid X = x)$ on classes $[K] := \{1, \cdots, K\}$ is calculated. Conformal prediction leverages an additional calibration step to guarantee a user-specified (marginal) coverage: by producing a prediction set $\mathcal{C}(x) \subseteq [K]$, it guarantees the true class of $x$ is included in $\mathcal{C}(x)$ with a user-chosen probability, when the calibration samples are exchangeable with the test samples.

The uncertainty typically manifests in two aspects in CP: (1) the efficiency of prediction sets; (2) the entropy of model predictions. For the former, $\mathcal{C}(x)$ with a small size is considered to have high efficiency, providing more certainty for decision-making processes. For the latter, entropy directly quantifies the level of prediction uncertainty. Simply consider the two prediction sets for a patient, {*Diabetes, Asthma*} with predictive probabilities 0.4 and 0.4, and {*Diabetes, Asthma, Stroke*} with predictive probabilities 0.6, 0.1 and 0.1. It would be difficult to compare the goodness of these two sets in terms of guiding a doctor to make decisions.

Recent progress in CP mainly focuses on the low-efficiency problem via introducing extra training on the base model, largely neglecting the important role of entropy. For example, Bellotti (2020) proposes the notion of conformal training and Stutz et al. (2022) simulate the conformal prediction process during training; this approach is further extended to graph-structure data (Huang et al., 2024b) by introducing a conformal adapter, which performs an additional conformal-aware training step based on the fixed base model. In this paper, we adopt the latter setting as the conformal adapter only needs the output distribution of the base model (as input), which is more akin to traditional CP (in the sense that it is decoupled from the base model), and thus has broader applications in practice. We refer to this emerging class of approaches as *conformal correction*. To be more concrete, given a base classifier $\widetilde{M}$, we can obtain a conformal adapter $\widehat{M}$, which takes $\tilde{\pi}(x)$ from $\widetilde{M}$ as input and outputs $\hat{\pi}(x)$, together with $\mathcal{C}(x)$ typically of a smaller size.

Our motivation is to have an in-depth understanding of the potential catch when a smaller $\mathcal{C}(x)$ is in place. We find that while the average size of $\mathcal{C}(x)$ may be smaller, the entropy of the prediction

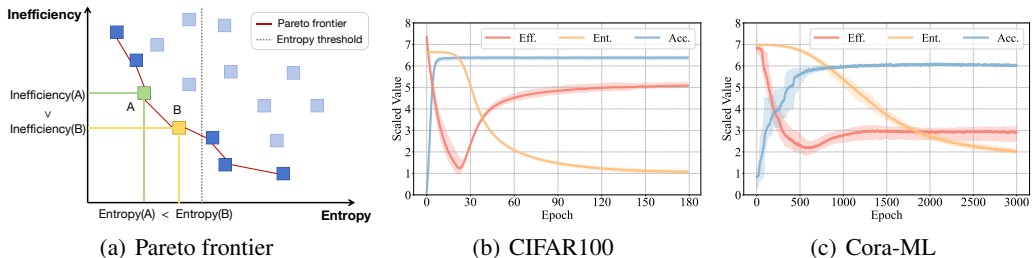

(a) Pareto frontier          (b) CIFAR100          (c) Cora-ML

Figure 1: Fig.(a) plots the Pareto frontier between inefficiency and entropy. For both of them, the lower, the better; Fig.(b) and (c) are the results of training only with $\mathcal{L}_{\text{class}}$ on CIFAR100 and Cora-ML, respectively. (b) and (c) depict the efficiency and entropy on the test set during the conformal correction, and there is a trade-off between them when the accuracy reaches the top.

$\hat{\pi}(x)$ also increases, indicating that the prediction becomes more *uncertain*, which is not ideal. Nevertheless, high efficiency should not sacrifice prediction entropy too much!

Indeed, our experiments show that, when conformal correction is applied on CIFAR100, the average size of CP sets is increased from 17.3 to 58.6 while the prediction entropy decreases from 6.3 to 1.1 (cf. Fig. 1(b) in Section 3). This indicates that a trade-off exists between the CP efficiency and the prediction entropy. We further confirm the finding by showing that, for APS (Romano et al., 2020), the expected size of CP sets can be upper-bounded by the negative entropy (plus some positive constant; cf. Theorem 3 for a precise account). This gives theoretical evidence that the efficiency of CP sets produced by APS may be at odds with the prediction entropy.

The trade-off between efficiency and entropy entails a Pareto perspective on CP, where different Pareto optima form a Pareto frontier as shown in Fig. 1(a). Conformal correction can thus be viewed as a traversal of the Pareto frontier. Technically, one can reduce the inefficiency significantly, but at the cost of an increased entropy, rendering such a reduction less meaningful. Instead, we argue that seeking for a better Pareto frontier is more crucial for conformal correction than simply adapting the trade-off. To this end, we propose a new method, i.e., **entropy-constrained conformal correction** (EC[3]) to ameliorate the trade-off by controlling the entropy of conformal adapters via focal loss (Mukhoti et al., 2020) and temperature scaling (Guo et al., 2017).

We conduct extensive experiments on computer vision (CV) and graph datasets to evaluate the effectiveness of EC[3]. The results show that our method can outperform the competitors by up to 34.4% in terms of efficiency given an entropy threshold; the qualitative analysis also indicates that our method can locate better Pareto optimality with strong control over the model entropy. Furthermore, when EC[3] is adapted to provide (stronger) conditional coverage, it can significantly improve, for instance, the class coverage from 0.77 to 0.83 (for the CV dataset) and from 0.74 to 0.85 (for the graph dataset), respectively.

To summarize, the main contributions of the paper are: (1) we identify a trade-off between efficiency and prediction entropy in CP, which has not been fully investigated before; (2) we propose a new conformal correction method based on entropy control to improve the efficiency of CP, the effectiveness of which is confirmed by extensive experiments.

## 2 PRELIMINARY

**Notations.** We focus on multiclass classification (with $K$ classes). Assume $D = \{(X_i, Y_i)\}_{i=1}^{n+1}$ of i.i.d. (or simply exchangeable) observations sampled from an (unknown) testing distribution $P_{XY}$. We denote the (oracle) conditional distribution $P_{Y|X}$ by $\pi_y(x) = P(Y = y \mid X = x)$. Furthermore, a black-box base classifier undergoes adapting to prescribe prediction $\hat{\pi}_y(x)$. The prediction entropy is defined as $H(\hat{\pi}(x)) := -\sum_{k=1}^{K} \hat{\pi}_k(x) \log \hat{\pi}_k(x)$.

**CP Framework.** Given $D_{\text{cal}} = \{(X_i, Y_i)\}_{i=1}^{n}$ as the calibration set and a user-defined miscoverage rate $\alpha \in (0, 1)$, CP typically proceeds in the following three steps:

*(1) Non-conformity score definition.* CP first heuristically defines a non-conformity score function $V(x, y)$, which indicates how the class $y$ conforms to the predictive result $\hat{\pi}(x) = [\hat{\pi}_1(x), \dots, \hat{\pi}_K(x)]$.

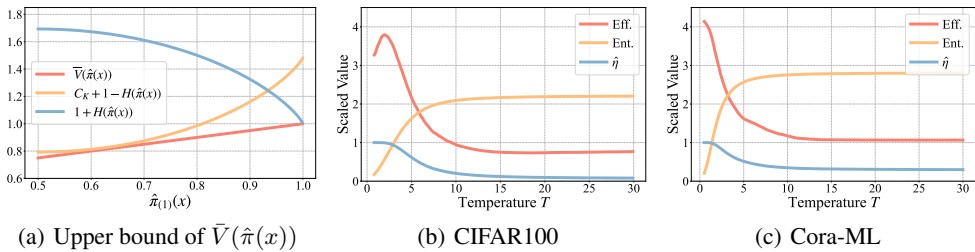

(a) Upper bound of $\bar{V}(\hat{\pi}(x))$      (b) CIFAR100      (c) Cora-ML

Figure 2: Fig. (a) is an illustration of Proposition 1 when $K = 2$, which demonstrates that the tight upper-bound of $\bar{V}(\hat{\pi}(x))$ consists of two pieces; Fig. (b) and (c) are efficiency and entropy curves of APS (cf. Section 2) on the test set after temperature scaling w.r.t. $T$ when $\alpha = 0.1$. The efficiency of APS is at odds with the entropy of model prediction in most cases.

For example, the non-conformity score $V(x, y)$ can be defined as the sum of the probabilities of all $K$ classes in $\hat{\pi}(x)$ except class $y$.

*(2) Uncertainty calibration.* CP then evaluates the non-conformity score for each data point $(X_i, Y_i) \in D_{\text{cal}}$, resulting in the non-conformity score set $\{V(X_i, Y_i)\}_{i=1}^n$. Subsequently, it sets a threshold $\hat{\eta}$ as its $(1 - \alpha)(1 + 1/n)$-quantile.

*(3) Prediction set construction.* For a new sample $X_{n+1}$, conformal prediction computes the corresponding prediction set by $\mathcal{C}(X_{n+1}) = \{y \in [K] \mid V(X_{n+1}, y) \leq \hat{\eta}\}$.

Traditional CP is model-agnostic, as it only requires prediction from the base model. Moreover, various non-conformity scores can be used to instantiate the framework (Romano et al., 2020; Angelopoulos et al., 2020). For instance, the Adaptive Prediction Set (APS) (Romano et al., 2020), the most classical adaptive conformal prediction approach, first sorts the predicted results in descending order, i.e., $\hat{\pi}_{(1)}(x) \geq \hat{\pi}_{(2)}(x) \geq \cdots \geq \hat{\pi}_{(k)}(x)$. The non-conformity score is then defined by the cumulative probabilities from the most likely class to the observed class $y$ in the calibration step, i.e., $V(x, y) = \sum_{i=1}^{y} \hat{\pi}_{(i)}(x)$.

**CP Evaluation.** The traditional methods focus on two dimensions for evaluating the quality of prediction sets, i.e., efficiency and coverage. (In)efficiency captures the average size of the prediction sets, i.e., $\mathbb{E}(|\mathcal{C}(X_{n+1})|)$; for inefficiency, the smaller, the better. For coverage, CP ensures the marginal coverage, viz., the true class $Y_{n+1}$ is in $\mathcal{C}(X_{n+1})$ with a probability of at least $1 - \alpha$, i.e.,

$$P(Y_{n+1} \in \mathcal{C}(X_{n+1})) \geq 1 - \alpha.$$

In certain cases, we also expect the conditional coverage to exceed $1 - \alpha$ for each $x$, i.e., $P(Y_{n+1} \in \mathcal{C}(x) \mid X_{n+1} = x) \geq 1 - \alpha$.

## 3 EFFICIENCY AND ENTROPY TRADE-OFF

In this section, we explore the trade-off between efficiency and entropy empirically and theoretically.

Existing work (Stutz et al., 2022; Huang et al., 2024b) has demonstrated that conformal correction can significantly reduce the sizes of CP sets. To this end, various sorting-smooth techniques (Blondel et al., 2020; Petersen et al., 2021) are employed to encode the size of CP sets in the loss function. The general optimization objective for conformal correction can be formulated as

$$\min \mathcal{L}_f := \mathcal{L}_{\text{class}} + \beta \cdot \mathcal{L}_{\text{ineff}}, \tag{1}$$

where $\mathcal{L}_{\text{class}}$ is the standard cross-entropy loss function for classification, $\mathcal{L}_{\text{ineff}}$ is the inefficiency loss function aiming to reduce the prediction set size, and $\beta$ is a hyperparameter.

**Empirical Observations.** Eq. (1) integrates inefficiency and classification losses by a weighted sum. However, a critical issue is whether the two losses, $\mathcal{L}_{\text{class}}$ and $\mathcal{L}_{\text{ineff}}$, can achieve their minima simultaneously. To explore this problem, we train a classifier only using $\mathcal{L}_{\text{class}}$ and plot the test curves of accuracy, efficiency, and entropy in Fig. 1(b) and 1(c). An interesting observation is that the accuracy and efficiency increase together while the entropy decreases during the initial training stage.

When the accuracy converges, as the entropy of the models drops—meaning the model becomes more certain—the efficiency decreases. This phenomenon suggests a trade-off between efficiency and entropy may exist when the model is fully trained.

**Theoretical Explanations.** We confirm the empirical observation by analyzing the impact of prediction entropy on CP efficiency in the context of APS. We first define the average non-conformity score

$$\bar{V}(\hat{\pi}(x)) = \frac{1}{K} \sum_{i=1}^{K} V(\hat{\pi}(x), i),$$

where $V(\hat{\pi}(x), i)$ refers to the APS non-conformity score of the $i$-th class. The average non-conformity score indicates the overall performance of the predictive result $\hat{\pi}(x)$ conforming to the class set $[K]$. Then, we establish a relationship between the average non-conformity score $\bar{V}(\hat{\pi}(x))$ and the prediction entropy $H(\hat{\pi}(x))$.

**Proposition 1** *For a given sample point $x$ and the corresponding predictive distribution $\hat{\pi}(x)$, the average non-conformity score is upper-bounded by the prediction entropy. Namely,*

$$\bar{V}(\hat{\pi}(x)) \leq \min(C_K + 1 - H(\hat{\pi}(x)), 1 + H(\hat{\pi}(x))),$$

*with constant $C_K := \log\left(\sum_{k=1}^{K} \exp(-\frac{k-1}{K})\right)$.*

The proof is given in Appendix A.1. Additionally, we use binary classification ($K = 2$) to illustrate this proposition by plotting the curves of $\bar{V}(\hat{\pi}(x))$ and its two upper bounds in Fig. 2(a). In this case, the entropy $H(\hat{\pi}(x))$ is determined by $\hat{\pi}_{(1)}(x)$ as $\hat{\pi}_{(2)}(x) = 1 - \hat{\pi}_{(1)}(x)$ and $\hat{\pi}_{(1)}(x) \geq \hat{\pi}_{(2)}(x)$. It can be observed that, with the decrease of entropy $H(\hat{\pi}(x))$, the tighter bound shifts from $C_K + 1 - H(\hat{\pi}(x))$ (the orange curve) to $1 + H(\hat{\pi}(x))$ (the blue curve).

Proposition 1 illustrates the relationship between the average non-conformity score and the entropy of model prediction. Moreover, the derived analysis is consistent with our empirical observation: when the entropy $H(\hat{\pi}(x))$ is sufficiently small, the average non-conformity is bounded by $1 + H(\hat{\pi}(x))$, allowing them to increase simultaneously. However, as $H(\hat{\pi}(x))$ becomes larger, $C_K + 1 - H(\hat{\pi}(x))$ becomes the tighter bound, which would prohibit the average non-conformity score from growing, leading to a trade-off in between.

Next, we extend the upper bound to the $(1 - \alpha)$-quantile $\hat{\eta}$.

**Proposition 2** *Given a sample subset $\mathcal{C}_{\hat{\eta}} := \{(X, Y) \mid V(X, Y) \geq \hat{\eta}\}$ in $\mathcal{D}$, the $(1 - \alpha)$-quantile $\hat{\eta}$ is upper bounded*

$$\hat{\eta} \leq \mathbb{E}[\bar{V}(X) \mid \mathcal{C}_{\hat{\eta}}] + C_{(\pi, K)} + \tau,$$

*with the probability at least $1 - \exp(-\frac{2\alpha\tau^2 n}{(1-\hat{\eta})^2})$, where $n$ is the size of the calibration set, $\tau$ is a positive constant, and the constant $C_{(\pi, K)} := \mathbb{E}[\sqrt{2(H(\pi(X)) + \log(K))} \mid \mathcal{C}_{\hat{\eta}}]$.*

The proof is given in Appendix A.2. Note that as $n$ increases, $1 - \exp(-\frac{2\alpha\tau^2 n}{(1-\hat{\eta})^2})$ tends to 1. Moreover, as the second term and third term of the upper bound are both constants, Proposition 2 effectively bounds $\hat{\eta}$ by the average non-conformity score $\bar{V}(X)$ from a sample subset $\mathcal{C}_{\hat{\eta}}$.

By combining Proposition 1 and Proposition 2, we can finally establish the trade-off between the expected size of conformal prediction sets $\mathbb{E}[|\mathcal{C}(X)|]$ and the entropy of model prediction.

**Theorem 3** *Let $\mu = \mathbb{P}\left(H(\hat{\pi}(X)) \geq \frac{1}{2}C_K \mid \mathcal{C}_{\hat{\eta}}\right)$. We have that*

$$\mathbb{E}[|\mathcal{C}(X)|] \leq \underbrace{K(1 - \alpha)(1 - 2\mu)\mathbb{E}[H(\hat{\pi}(x)) \mid \mathcal{C}_{\hat{\eta}}]}_{(\star)} + \mathcal{O}(K).$$

The proof is provided in Appendix A.3.

**Remark 4** By Theorem 3, we can see that, when $\mu \geq \frac{1}{2}$ (i.e., the entropy of a majority of $x$ in the subset $\mathcal{C}_{\hat{\eta}}$ is greater than $\frac{1}{2}C_K$), $1 - 2\mu < 0$ holds and so does term $(\star)$, which entails that $\mathbb{E}(|\mathcal{C}(X)|)$ is at odds with the expected entropy $\mathbb{E}[H(\hat{\pi}(x)) \mid \mathcal{C}_{\hat{\eta}}]$. Otherwise, the term $(\star)$ is positive, allowing a potential synergy between efficiency and entropy.

Intuitively, for APS, the trade-off between efficiency and entropy will be present when the entropy is sufficiently large (roughly, greater than $\frac{1}{2}C_K$). It is not hard to see that $\frac{1}{2}C_K$ is a monotonically increasing function in $K$. This suggests that, when $K$ is relatively small, the interval $[0, \frac{1}{2}C_K]$ is narrow, and thus the trade-off will be largely dominating.

# 4 CONFORMAL CORRECTION METHODS

In this section, we present a new method, EC$^3$, for conformal correction. In general, EC$^3$ is based on Section 3, searching for better Pareto optima via controlling the entropy of model predictions. Then, we directly utilize temperature scaling to explore the Pareto frontier, and extend EC$^3$ to improve the user-specified conditional coverage.

## 4.1 ENTROPY-CONSTRAINED CONFORMAL CORRECTION (EC$^3$)

As discussed in Section 3, there is a fundamental trade-off between conformal efficiency and prediction entropy. A natural way to search for the Pareto frontier is to introduce a positive entropy term into Eq. (1), and balance it with $\mathcal{L}_{\text{ineff}}$. However, in our case, the cross-entropy loss $\mathcal{L}_{\text{class}}$ already implicitly enforces entropy reduction.[1] Furthermore, we observe that directly optimizing the original training loss often results in a rapid decline in efficiency, leading to low-entropy solutions with poor efficiency (e.g., after the 30-th epoch in Fig. 1(b)). Therefore, we add a negative entropy term into Eq. (1) to counter the rapid decline in efficiency, which enables a more fine-grained control of entropy during conformal correction, viz.,

$$\min \mathcal{L}_f = \mathcal{L}_{\text{class}} + \beta \cdot \mathcal{L}_{\text{ineff}} - \gamma \cdot H(\hat{\pi}(x)), \tag{2}$$

where $\gamma \geq 0$ is a hyperparameter controlling the weight of the entropy term.

There are three competing optimization objectives in Eq. (2), making the optimization challenging. Fortunately, the following inequality for $\mathcal{L}_{\text{class}}$ and $H(\hat{\pi}(x))$ holds (Mukhoti et al., 2020):

$$\mathcal{L}_{\text{focal}} \geq \text{KL}(\pi(x)||\hat{\pi}(x)) - \gamma \cdot H(\hat{\pi}(x)),$$

where $\mathcal{L}_{\text{focal}} = -\sum_{k=1}^{K}(1-\hat{\pi}_k(x))^\gamma \pi_k(x) \log \hat{\pi}_k(x)$ is the form of focal loss (Mukhoti et al., 2020) and $\text{KL}(\pi(x)||\hat{\pi}(x))$ is the KL-divergence between the ground-truth distribution $\pi(x)$ and the model prediction $\hat{\pi}(x)$. Since $\text{KL}(\pi(x)||\hat{\pi}(x))$ can be reduced to the cross-entropy, this inequality allows us to directly optimize the upper bound for $\mathcal{L}_{\text{class}} - \gamma \cdot H(\hat{\pi}(x))$, i.e., $\mathcal{L}_{\text{focal}}$. Thus, we rewrite the objective in Eq. (2) as

$$\min \mathcal{L}_f = \mathcal{L}_{\text{focal}} + \beta \cdot \mathcal{L}_{\text{ineff}}. \tag{3}$$

Compared with Eq. (1), the above objective enjoys two advantages. From the view of multi-objective optimization, the proposed minimization objective can flexibly adjust the trade-off between the CP efficiency and the entropy of prediction by controlling $\gamma$. Specifically, if we prefer CP efficiency over entropy, $\gamma$ should be augmented to increase the efficiency at the cost of entropy. Otherwise, we should lower the value of $\gamma$. When $\gamma = 0$, Eq. (3) degrades into Eq. (1).

Additionally, in contrast to directly penalizing the entropy, focal loss presents a more flexible approach to balance classification loss and entropy regularization through the coefficient $(1-\hat{\pi}_k(x))^\gamma$. Specifically, the focal loss can effectively control the strength of the classification loss based on the sharpness of the model prediction $\hat{\pi}(x)$. This adaptation facilitates locating better solutions in the trade-off between entropy and conformal efficiency.

**Pareto Frontier Exploration via Temperature Scaling.** When a better Pareto optimum is achieved by Eq. (3), we can next traverse the Pareto frontier from this Pareto optimum via temperature scaling—a common trick used in model calibration (Guo et al., 2017)—to flexibly regulate the entropy. Specifically, it rephrases the softmax function as

$$\hat{\pi}_i(x) = \frac{\exp(\hat{\pi}_i(x)/T)}{\sum_{j=1}^{K} \exp(\hat{\pi}_j(x)/T)},$$

---

[1]Minimizing cross-entropy loss essentially encourages the predicted distribution to approximate a sharp distribution (e.g., the one-hot label vector), rendering a trained model with low entropy, which is also known as the over-confident problem (Guo et al., 2017).

Table 1: The efficiency results of conformal correction methods on CV datasets when $\alpha = 0.1$. The results are the average of five runs of the pre-trained model, each with 100 runs of conformal splits on CV datasets. The best results are in shadow. The proposed EC$^3$ achieves the best efficiency performance and maintains the marginal coverage.

| DATASET | MODEL | CP | | CONFTR | | EC$^3$ | |
|---|---|---|---|---|---|---|---|
| | | COVERAGE | EFFICIENCY | COVERAGE | EFFICIENCY | COVERAGE | EFFICIENCY |
| CIFAR10 | RESNET56 | $0.90_{\pm.00}$ | $5.41_{\pm.11}$ | $0.90_{\pm.00}$ | $1.31_{\pm.15}$ | $0.90_{\pm.00}$ | $1.23_{\pm.06}$ |
| | PRERESNET110 | $0.90_{\pm.00}$ | $5.54_{\pm.07}$ | $0.90_{\pm.00}$ | $1.25_{\pm.16}$ | $0.90_{\pm.00}$ | $1.18_{\pm.04}$ |
| | DENSENET100 | $0.90_{\pm.00}$ | $5.52_{\pm.03}$ | $0.90_{\pm.00}$ | $1.30_{\pm.15}$ | $0.90_{\pm.00}$ | $1.09_{\pm.04}$ |
| CIFAR100 | RESNET56 | $0.90_{\pm.00}$ | $23.06_{\pm.66}$ | $0.90_{\pm.00}$ | $19.83_{\pm1.94}$ | $0.90_{\pm.00}$ | $18.05_{\pm2.71}$ |
| | PRERESNET110 | $0.90_{\pm.00}$ | $25.93_{\pm.37}$ | $0.90_{\pm.00}$ | $17.62_{\pm1.98}$ | $0.90_{\pm.00}$ | $15.27_{\pm1.27}$ |
| | DENSENET100 | $0.90_{\pm.00}$ | $30.00_{\pm2.44}$ | $0.90_{\pm.00}$ | $13.29_{\pm1.61}$ | $0.90_{\pm.00}$ | $10.87_{\pm1.42}$ |

where $T > 0$ is the temperature controlling the uncertainty of models. With $T$ increasing, the prediction entropy $H(\hat{\pi}(x))$ becomes larger. Practically, temperature scaling is based on the grid search, which is simple and convenient to use.

Note that one can directly use temperature scaling to adapt the trade-off between efficiency and entropy. Fig. 2 plots the results of APS over temperature $T$ on CIFAR100 and Cora-ML datasets.

Fig. 2(b) depicts the result of the CIFAR100 dataset. We observe an initial concurrent increase in both efficiency and entropy as $T$ rises. However, as $T$ continues to increase, a trade-off emerges between these two metrics: efficiency decreases while entropy increases. This observation aligns with Theorem 3. That is, when entropy is relatively low (corresponding to lower values of $T$), efficiency is upper-bounded by the entropy; conversely, as entropy increases (with higher values of $T$), efficiency is upper-bounded by the *negative* entropy (plus some positive constant).

Fig. 2(c) depicts the result of the Cora-ML dataset. In contrast, only the trade-off between the efficiency and entropy can be observed. This is because the number of classes therein is significantly smaller than that in the CIFAR100 dataset, which is consistent with Remark 4. In addition to the results on CIFAR100 and Cora-ML datasets, we relegate the rest to Appendix B.5.

Nevertheless, directly using temperature scaling without the objective in Eq. (3) will lead to a suboptimal Pareto frontier (cf. Fig. 7 in Appendix B.8).

## 4.2 EXTENSIONS TO CONDITIONAL COVERAGE

Recall from Section 2 that conditional coverage is stronger than marginal coverage. The flexibility of the EC$^3$ approach allows us to adjust user-specified conditional coverage adaptively. Take the class conditional coverage (i.e., the coverage of the sample subsets with the same true class (Zargarbashi et al., 2023)) as an example. We define the following class conditional coverage loss function for each class $k$,

$$\mathcal{L}_k := -\frac{1}{|D_{\text{cal}}^k|} \sum_{(x,y) \in D_{\text{cal}}^k} \mathbb{I}(y \in \mathcal{C}(x)),$$

where $D_{\text{cal}}^k := \{(x,y) \in D_{\text{cal}} \mid y = k\}$ and $\mathbb{I}(\cdot)$ is the indicator function. We then obtain a new minimization objective to improve the class conditional coverage during conformal correction,

$$\min \mathcal{L}_f = \mathcal{L}_{\text{focal}} + \beta \cdot \mathcal{L}_{\text{ineff}} - \frac{1}{K} \sum_{k=1}^{K} \mathcal{L}_k.$$

We refer to this method as EC$^3$ (Cond) which will be evaluated in Section 5.2.

## 5 EXPERIMENTS

### 5.1 EXPERIMENTAL SETUP

**Datasets.** We conduct main experiments on five datasets, including CIFAR10, CIFAR100 (Krizhevsky et al., 2009), Cora-ML (McCallum et al., 2000), CS (Shchur et al., 2018), and Photos (McAuley et al.,

Table 2: The efficiency results of conformal correction methods on graph datasets when $\alpha = 0.1$. The results are the average of five runs of the pre-trained model, each with 100 runs of conformal splits on graph datasets. The best results are in shadow. The proposed EC$^3$ achieves the best efficiency performance and maintains the marginal coverage.

| DATASET | MODEL | CP | | CF-GNN | | EC$^3$ | |
|---|---|---|---|---|---|---|---|
| | | COVERAGE | EFFICIENCY | COVERAGE | EFFICIENCY | COVERAGE | EFFICIENCY |
| CORA-ML | GCN | $0.90_{\pm.00}$ | $4.00_{\pm.19}$ | $0.90_{\pm.00}$ | $1.85_{\pm.26}$ | $0.90_{\pm.00}$ | $1.50_{\pm.13}$ |
| | GAT | $0.90_{\pm.00}$ | $3.92_{\pm.13}$ | $0.90_{\pm.00}$ | $1.94_{\pm.39}$ | $0.90_{\pm.00}$ | $1.68_{\pm.13}$ |
| | SGC | $0.90_{\pm.00}$ | $4.01_{\pm.13}$ | $0.90_{\pm.00}$ | $1.81_{\pm.26}$ | $0.90_{\pm.00}$ | $1.58_{\pm.09}$ |
| CS | GCN | $0.90_{\pm.00}$ | $8.37_{\pm.22}$ | $0.90_{\pm.00}$ | $4.45_{\pm.38}$ | $0.90_{\pm.00}$ | $3.13_{\pm.21}$ |
| | GAT | $0.90_{\pm.00}$ | $6.92_{\pm.36}$ | $0.90_{\pm.00}$ | $4.47_{\pm.50}$ | $0.90_{\pm.00}$ | $3.03_{\pm.36}$ |
| | SGC | $0.90_{\pm.00}$ | $8.37_{\pm.26}$ | $0.90_{\pm.00}$ | $4.36_{\pm.33}$ | $0.90_{\pm.00}$ | $3.13_{\pm.25}$ |
| PHOTOS | GCN | $0.90_{\pm.00}$ | $4.00_{\pm.14}$ | $0.90_{\pm.00}$ | $2.07_{\pm.38}$ | $0.90_{\pm.00}$ | $1.54_{\pm.14}$ |
| | GAT | $0.90_{\pm.00}$ | $2.36_{\pm.24}$ | $0.90_{\pm.00}$ | $2.69_{\pm.30}$ | $0.90_{\pm.00}$ | $2.13_{\pm.08}$ |
| | SGC | $0.90_{\pm.00}$ | $4.00_{\pm.06}$ | $0.90_{\pm.00}$ | $2.17_{\pm.28}$ | $0.90_{\pm.00}$ | $1.53_{\pm.06}$ |

2015), as detailed in Appendix B.1. Following Huang et al. (2024b), we randomly split each dataset into the training set $D_{\text{train}}$, validation set $D_{\text{valid}}$, calibration set $D_{\text{cal}}$ and testing set $D_{\text{test}}$ with the ratio 2:1:4:3. We perform 100 random splits of calibration/testing sets, and report the average results and standard deviations to suppress randomness. Additionally, the information of base and adapter models are introduced in Appendix B.1.

**Baselines.** We select the state-of-the-art methods, i.e., ConfTr (Stutz et al., 2022) and CF-GNN (Huang et al., 2024b) from CV and graph domains, respectively. Note that while CF-GNN strictly follows the conformal correction framework, ConfTr was initially proposed to retrain the base model. In our experiments, we adapt ConfTr to the conformal correction framework (i.e., applying its optimization objective as an adapter after the base model is trained); further performance improvements are observed.

**Evaluation Metrics.** For efficiency, we use the standard *average size* of CP sets as the metric. For marginal coverage, we can directly compute its value. For conditional coverage, we consider WSC and SSCV metrics (Romano et al., 2020; Angelopoulos et al., 2020). All these metrics are widely adopted by existing work.

**Implementations.**[2] To construct the CP sets, we consider both APS (Romano et al., 2020) and RAPS (Angelopoulos et al., 2020). We report the APS results in the main body of the paper, and the results on RAPS are included in Appendix B.6.

We set the threshold of the prediction entropy to be $(1 - \epsilon) \exp(\log K)$, and use $\epsilon = 1/4$ by default. We also set the miscoverage rate $\alpha = 0.1$, hyperparameter $\beta = 0.1$ and $\gamma = 4$ by default. All the experiments are carried out on NVIDIA GeForce RTX 3090. More implementation details, such as the hyperparameters of base models and conformal adapters, are presented in Appendix B.1.

## 5.2 EXPERIMENTAL RESULTS

**Efficiency Comparison.** Give an entropy threshold as mentioned in Section 5.1, we first compare the marginal coverage and efficiency of training-based conformal correction methods when $\alpha = 0.1$, the results of which are given in Table 1 and Table 2. We also report the comparison results when $\alpha = 0.2$ (in Appendix B.2), the entropy results (Appendix B.3) and the accuracy results (in Appendix B.4).

On the CV datasets, the proposed EC$^3$ method performs better than the baseline ConfTr in terms of efficiency over all three pre-trained models on both datasets, while it keeps the marginal coverage at the same time (cf. Table 1). For example, EC$^3$ is 18.2% more efficient than ConfTr for the pre-trained model DenseNet100 on CIFAR100. Similarly, EC$^3$ achieves significant efficiency improvements from 12.7% to 34.4% on the graph datasets, as shown in Table 2.

---

[2]An implementation of our conformal correction approach can be accessed at the following anonymous link: https://github.com/Anonymity23143/Conformal-Correction.

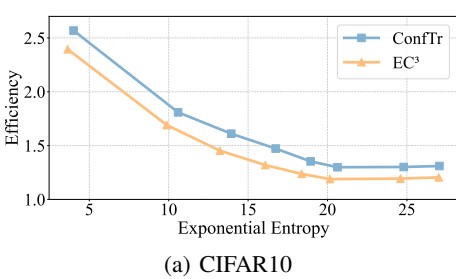 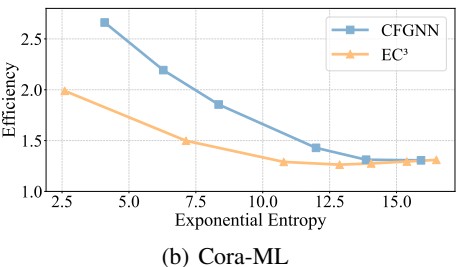

(a) CIFAR10  (b) Cora-ML

Figure 3: Pareto optima of different conformal correction methods. Compared with baselines, the proposed $EC^3$ obtains the best Pareto frontier via achieving a better balance between efficiency and entropy on both CIFAR10 and Cora-ML.

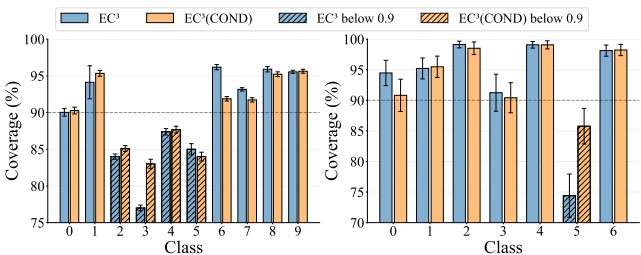

Figure 4: Class conditional coverage results on CIFAR10 (left) and Cora-ML (right). The class coverage below 0.9 is in shadow. Our method $EC^3$ (Cond) increases most of the class coverages below 0.9.

| MODEL | CIFAR10 | | CORA-ML | |
|---|---|---|---|---|
| | L1 | L2 | L1 | L2 |
| $EC^3$ | 0.25 | 0.15 | 0.16 | 0.16 |
| $EC^3$ (COND) | 0.21 | 0.11 | 0.04 | 0.04 |
| IMP. | 18% | 26% | 73% | 73% |

Table 3: Distances between the class coverage below 0.9 and the target coverage 0.9 with L1-norm and L2-norm in the left figures. $EC^3$ (Cond) improves the class conditional coverage successfully by up to 73%.

The significant efficiency improvement can be attributed to the theoretical analysis of the efficiency-entropy tradeoff which $EC^3$ is built on. In particular, the explicit modeling of entropy enables $EC^3$ to achieve a better balance between efficiency and entropy within an acceptable entropy range.

**Pareto Frontiers.** We explore the Pareto frontier of efficiency and entropy for all conformal correction methods with temperature scaling. We can directly control entropy by adjusting the temperature $T$ and select sufficient values of $T$ to cover the entropy range.

The results of the Pareto frontier are shown in Fig. 3. It can be observed that $EC^3$ (the orange curve) achieves a better Pareto frontier (i.e., lower efficiency given the same entropy) than other conformal correction methods (the blue curve) in both Fig. 3(a) and Fig. 3(b). The contrast of Pareto frontiers further confirms the positive impact of the entropy control on the efficiency-entropy trade-off, in terms of seeking better Pareto optima.

**Conditional Coverage.** We present the conditional coverage results before and after conformal correction by the metrics WSC and SSCV in Appendix B.7.

Next, we evaluate the effectiveness of the conditional conformal correction presented in Section 4.2. In Fig. 4, we plot the histogram of the coverage of different classes, without and with the conditional conformal correction, i.e., $EC^3$ and $EC^3$ (Cond). As mentioned in Section 2, class conditional coverage requires that the coverage of each class is greater than $1 - \alpha$. Hence, we only need to examine the classes whose coverage is below $1 - \alpha$. In Fig. 4, we can observe that the conditional conformal correction improves most of the class coverages below 0.9. In particular, the lowest class coverages are increased from 0.77 to 0.83 and from 0.74 to 0.85, respectively. Moreover, we compute the distance between the class coverage below 0.9 and the target coverage of 0.9 with L1-norm and L2-norm in Table 3. The results show that $EC^3$ (Cond) significantly reduces such distances by the ratio from 18% to 73%.

**Additional Results.** Fig. 7 in Appendix B.8 summarizes additional experimental results about the sensitivity of temperature $T$ and hyperparameter $\gamma$. Moreover, Appendix B.9 showcases the empirical performance of our $EC^3$ extension for the question answering task on LLMs; see Fig. 6(b) and Table 11 for more details.

## 6 RELATED WORK

**Conformal Prediction.** Uncertainty Quantification (UQ) (Abdar et al., 2021) aims to provide calibrated uncertainty estimates for machine learning models, enabling reliable decision-making in critical applications. UQ has been widely studied in both classification and regression, where typical methods include Bayesian methods (Gal & Ghahramani, 2016), confidence calibration (Guo et al., 2017), and model-agnostic frameworks that construct uncertainty intervals (Romano et al., 2019). However, most of these methods fail to provide rigorous statistical guarantees regarding coverage, especially in non-i.i.d. settings.

CP, as a UQ method, distinguishes itself in providing guaranteed coverage, regardless of the underlying model or data distribution. It has been applied to diverse domains, including image classification (Sadinle et al., 2019), object detection (Teng et al., 2023), and large language models (Kumar et al., 2023). Most CP methods rely on splitting the dataset into a training set and a held-out calibration set to estimate non-conformity scores, as proposed in split conformal prediction (Lei et al., 2015). Other extensions, such as jackknife methods (Barber et al., 2021) and cross-validation-based approaches (Vovk, 2015), can further enhance CP's flexibility and applicability.

Some work has focused on improving the efficiency and adaptability of CP by refining non-conformity scores. Adaptive Prediction Sets (APS) (Romano et al., 2020) introduced a score function that accumulates sorted softmax probabilities; Regularized Adaptive Prediction Sets (RAPS) (Angelopoulos et al., 2020) extended APS by adding penalties to tail classes and Sorted Adaptive Prediction Sets (SAPS) (Huang et al., 2024a) substituted probability values with sort orders, both resulting in more efficient prediction sets with minimal computational overhead.

**Conformal Correction.** These techniques aim to improve the performance of CP by modifying the model's output via extra training. Although computationally intensive, model correction represents a significant advancement in enhancing CP's performance. For instance, ConfTr (Stutz et al., 2022) proposes non-conformity loss functions designed to align scores with a uniform distribution. Similar approaches have also demonstrated effectiveness in graph-structured data. CF-GNN (Huang et al., 2024b), for example, introduces an additional correction model that utilizes the graph's topology to adjust the model's output.

Our approach also optimizes CP during training by modifying the model's output. Furthermore, our work investigates the relationship between the entropy and efficiency of CP. While the experimental results of ConfTS (Xi et al., 2024) corroborate part of this relationship, we provide a comprehensive theoretical framework. By introducing a novel loss function, we achieve superior efficiency and prediction performance compared to existing correction-based optimization methods. Correia et al. (2024) provide a lower bound of the expected size of the conformal prediction sets (i.e., inefficiency). In contrast, our work provides an upper bound, which is more important for improving efficiency. We mention that, in addition to efficiency, some work has also focused on conditional coverage of CP through model correction (Einbinder et al., 2022; Kiyani et al., 2024) and discussed the stability of conformal training (Noorani et al., 2025).

## 7 CONCLUSION

In this paper, we have demonstrated that a decrease in the inefficiency of CP is often accompanied by an increase in the prediction entropy during conformal correction. We have also provided a theoretical analysis explaining this phenomenon. Both lead to the conclusion that CP efficiency may be at odds with the prediction entropy in most cases. The trade-off between them hints at a Pareto optimality view of conformal correction, for which we have proposed a new method $EC^3$. Experiments on both CV and graph datasets showcase that it outperforms the existing baselines.

**Limitations.** In this work, our theoretical analysis and methods mainly target adaptive conformal prediction, which are the mainstream conformal methods for classification (Smith, 2024). They involve numerous non-conformity scores, which uniquely take the conditional coverage into account (Romano et al., 2020; Angelopoulos et al., 2020; Fontana et al., 2023). Additionally, the proposed method may slightly sacrifice the accuracy of models, similar to current training-based conformal correction approaches (Stutz et al., 2022; Huang et al., 2024b).

## REPRODUCIBILITY STATEMENT

To ensure the reproducibility of our work, we have made the source code publicly available via an anonymous repository. Appendix B.1 provides a comprehensive description of all experimental details, including dataset partitions, hyperparameter configurations, and base model specifications. We are confident that these resources are sufficient to reproduce all findings reported in this paper.

## USAGE OF LLMS

We employed Large Language Models (LLMs) solely as writing assistants for polishing the manuscript's language, with the goal of improving clarity, grammar, and readability. Their use was restricted to this editorial role. LLMs were not involved in any core research activities—including idea generation, experimental design, data analysis, results interpretation, or scientific content creation. All intellectual contributions, methodologies, findings, and conclusions entirely belong to the authors.

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

## A    TECHNICAL PROOFS FOR THEORETICAL RESULTS

### A.1    PROOF OF PROPOSTION 1

**Proof**  Recall that the non-conformity scores of APS are defined by $V(x, y) = \hat{\pi}_{(1)}(x) + \cdots + \hat{\pi}_{(y)}(x)$, where $\hat{\pi}_{(1)}(x) \geq \cdots \geq \hat{\pi}_{(K)}(x)$ are (ordered) probabilities of the model prediction. Hence, we have

$$
\begin{aligned}
\bar{V}(\hat{\pi}(x)) &= \frac{1}{K} \sum_{i=1}^{K} V(x, i) \\
&= \sum_{k=1}^{K} \frac{1}{K} (\hat{\pi}_{(1)}(x) + \cdots + \hat{\pi}_{(k)}(x)) \\
&= \hat{\pi}_{(1)}(x) + \frac{K-1}{K} \hat{\pi}_{(2)}(x) + \cdots + \frac{1}{K} \hat{\pi}_{(K)}(x).
\end{aligned}
\tag{4}
$$

Consider the function $F_1(\hat{\pi}) := \bar{V}(\hat{\pi}(x)) - H(\hat{\pi}(x))$, we can find that $F_1(\hat{\pi})$ is a convex function w.r.t. $\hat{\pi}(x)$, since the negative entropy is convex and the average non-conformity score is linear of $\hat{\pi}(x)$. Therefore, the upper bound of $F_1(\hat{\pi})$ is achieved at the boundary of constraints $\{\hat{\pi}_{(1)}(x) \geq \cdots \geq \hat{\pi}_{(K)}(x), \hat{\pi}_{(1)}(x) + \cdots + \hat{\pi}_{(K)}(x) = 1\}$. Furthermore, it can be observed that $\bar{V}(\hat{\pi}(x)))$ achieves the maximum value 1 and $-H(\hat{\pi}) = \sum_{k=1}^{K} \hat{\pi}_{(k)} \log \hat{\pi}_{(k)}$ reaches the maximum value 0 simultaneously when $\hat{\pi}_{(1)}(x) = 1$ and $\hat{\pi}_{(2)}(x) = \cdots = \hat{\pi}_{(K)} = 0$. Therefore, we have $F_1(\hat{\pi}) \leq 1$, which implies $\bar{V}(\hat{\pi}(x)) \leq 1 + H(\hat{\pi}(x))$.

We also define the function $F_2(\hat{\pi}) := \bar{V}(\hat{\pi}(x)) + H(\hat{\pi}(x))$. Similarly, we can find that $F_2(\hat{\pi}(x))$ is concave w.r.t. $\hat{\pi}(x)$. To analyze the upper bound of $F_2(\hat{\pi})$, we formulate the following optimization problem:

$$
\begin{aligned}
\max_{\pi} \quad & \bar{V}(\hat{\pi}(x)) + H(\hat{\pi}(x)) \\
\text{s.t.} \quad & \sum_{k=1}^{K} \hat{\pi}_{(k)}(x) = 1, \\
& \hat{\pi}_{(k)}(x) \geq \hat{\pi}_{(k+1)}(x) \quad k = 0, 1, \ldots, K-1,
\end{aligned}
$$

We temporarily drop the inequality constraints and compute the Lagrangian as

$$
\mathcal{L}_F(\hat{\pi}) = H(\hat{\pi}(x)) + \bar{V}(\hat{\pi}(x)) - \lambda(\sum_{k=1}^{K} \hat{\pi}_{(k)}(x) - 1),
$$

where $\lambda$ is the Lagrangian multiplier. Vanishing the partial derivatives, we obtain

$$
\frac{\partial \mathcal{L}_F(\hat{\pi})}{\partial \hat{\pi}_{(k)}(x)} = \log \hat{\pi}_{(k)}(x) + \frac{k-1}{K} - \lambda = 0.
$$

Hence, we have

$$
\hat{\pi}_{(k)}(x) = \exp(\lambda - \frac{k-1}{K}).
$$

Note that $\hat{\pi}_{(k)}(x) \geq \hat{\pi}_{(k+1)}(x)$ also holds for $i = 0, 1, \ldots, t-1$. Using the equation $\sum_{k=1}^{K} \hat{\pi}_{(k)}(x) = 1$, we can derive that

$$
\exp(\lambda) = \frac{1}{\sum_{k=1}^{K} \exp(-\frac{k-1}{K})}.
$$

In other words, the optimal solution is the output of the Softmax function for logits $[0, -\frac{1}{K}, \ldots, -\frac{K-1}{K}]$. By substituting the optimal solution, we have the maximum value of $F_2(\hat{\pi}(x))$:

$$F_2(\hat{\pi}(x)) \leq -\sum_{k=1}^{K}(\lambda - \frac{k-1}{K})\exp(\lambda - \frac{k-1}{K}) + (1 - \frac{k-1}{K})\exp(\lambda - \frac{k-1}{K})$$

$$= (-\lambda + 1)\sum_{k=1}^{K}\exp(\lambda - \frac{k-1}{K}) \tag{5}$$

$$= -\lambda + 1 = \log\big(\sum_{k=1}^{K}\exp(-\frac{k-1}{K})\big) + 1.$$

Therefore, we have

$$\bar{V}(\hat{\pi}(x)) \leq \log\big(\sum_{k=1}^{K}\exp(-\frac{k-1}{K})\big) + 1 - H(\hat{\pi}(x)).$$

Putting the two upper bounds together, we have

$$\bar{V}(\hat{\pi}(x)) \leq \min(C_K + 1 - H(\hat{\pi}(x)), 1 + H(\hat{\pi}(x))),$$

where $C_K = \log\big(\sum_{k=1}^{K}\exp(-\frac{k-1}{K})\big)$. Furthermore, we can see that the first upper bound strictly holds only when $H(\hat{\pi}(x)) \geq \frac{1}{2}C_K$. We complete the proof. $\square$

We plot the function image of $\frac{1}{2}C_K$ using Wolfram and find that $\frac{1}{2}C_K$ monotonically increases as $K$ rising. Thus, $C_K + 1 - H(\hat{\pi}(x))$ is a better upper bound than $1 + H(\hat{\pi}(x))$ in most cases, when $K$ is relatively small.

### A.2 PROOF OF PROPOSITION 2

**Proof** For a given sample point $x$ and its oracle distribution $\pi(x)$, we have

$$\mathbb{E}[V(X,Y) \mid X = x] = \sum_{k=1}^{K}\pi_{(k)}(x)\cdot\sum_{j=1}^{k}\hat{\pi}_{(j)}(x) \tag{6}$$

Next, we bridge $\bar{V}(x)$ and $\mathbb{E}[V(X,Y) \mid X = x]$. We can obtain that

$$|\mathbb{E}[V(X,Y) \mid X = x] - \bar{V}(x)| = \big|\sum_{k=1}^{K}\big(\pi_{(k)}(x) - \frac{1}{K}\big)\cdot\sum_{j=1}^{k}\hat{\pi}_{(j)}(x)\big|$$

$$\leq \sum_{k=1}^{K}\big|\pi_{(k)}(x) - \frac{1}{K}\big|\cdot\max\big\{\sum_{j=1}^{k}\hat{\pi}_{(j)}(x), k = 1, \ldots, K\big\}$$

$$= 2\delta_{\text{TVD}}(\pi(x), \mathcal{U}(K)),$$

where the inequation is derived by the Hölder inequality, and $\delta_{\text{TVD}}(\cdot,\cdot)$ refers to the total variation distance.

Using the Pinsker inequality, we have

$$\delta_{\text{TVD}}(\pi(x), \mathcal{U}(K)) \leq \sqrt{\frac{1}{2}\delta_{\text{KL}}(\pi(x)\|\mathcal{U}(K))} = \sqrt{\frac{1}{2}\big(H(\pi(x)) + \log(K)\big)}$$

Putting together, we obtain

$$|\mathbb{E}[V(X,Y) \mid X = x] - \bar{V}(x)| \leq \sqrt{2\big(H(\pi(x)) + \log(K)\big)}.$$

Next, we define $\mathcal{C}_{\hat{\eta}} := \{(X,Y) \mid V(X,Y) \geq \hat{\eta}\}$, and seek for the upper bound of $\mathbb{E}(V(X,Y) \mid \mathcal{C}_{\hat{\eta}})$ from the above result:

$$\mathbb{E}[V(X,Y) \mid \mathcal{C}_{\hat{\eta}}] \leq \mathbb{E}[\bar{V}(X) \mid \mathcal{C}_{\hat{\eta}}] + \mathbb{E}[\sqrt{2\big(H(\pi(X)) + \log(K)\big)} \mid \mathcal{C}_{\hat{\eta}}].$$

By Hoeffding's inequality and $V(X, Y) \in [\hat{\eta}, 1]$, and $|\mathcal{C}_{\hat{\eta}}| = \alpha n$, we have

$$\mathbb{P}\big(\frac{1}{n\alpha} \sum_{(X,Y) \in \mathcal{C}_{\hat{\eta}}} V(X,Y) - \mathbb{E}[V(X,Y) \mid \mathcal{C}_{\hat{\eta}}] \geq \tau\big) \leq \exp\left(-\frac{2\alpha\tau^2 n}{(1-\hat{\eta})^2}\right),$$

where $\tau$ is an arbitrary positive constant. Since we have $\sum_{(X,Y) \in \mathcal{C}_{\hat{\eta}}} V(X,Y) \geq n\alpha\hat{\eta}$, we can derive

$$\exp(-\frac{2\alpha\tau^2 n}{(1-\hat{\eta})^2}) \geq \mathbb{P}\big(\frac{1}{n\alpha} \sum_{(X,Y) \in \mathcal{C}_{\hat{\eta}}} V(X,Y) - \mathbb{E}[V(X,Y) \mid \mathcal{C}_{\hat{\eta}}] \geq \tau\big)$$

$$\geq \mathbb{P}(\hat{\eta} - \mathbb{E}[V(X,Y) \mid \mathcal{C}_{\hat{\eta}}] > \tau)$$

Substituting the bound of $\mathbb{E}[V(X,Y) \mid \mathcal{C}_{\hat{\eta}}]$ in, we have

$$\exp(-\frac{2\alpha\tau^2 n}{(1-\hat{\eta})^2}) \geq \mathbb{P}(\hat{\eta} > \mathbb{E}[V(X,Y) \mid \mathcal{C}_{\hat{\eta}}] + \tau)$$

$$\geq \mathbb{P}(\hat{\eta} > \mathbb{E}[\bar{V}(X) \mid \mathcal{C}_{\hat{\eta}}] + \mathbb{E}[\sqrt{2\big(H(\pi(X)) + \log(K)\big)} \mid \mathcal{C}_{\hat{\eta}}] + \tau),$$

which completes the proof. $\qquad\square$

### A.3  PROOF OF THEOREM 3

**Proof**  According to the definition of $\mathcal{C}(x)$ in terms of APS, we first have

$$|\mathcal{C}(x)| = S(\hat{\pi}(x), \hat{\eta}) = \sum_{k=1}^{K} u(\hat{\eta} - V(x,k)),$$

where $u(a)$ is an unit step function, i.e., if $a \geq 0$, $u(a) = 1$; otherwise, $u(a) = 0$. Since $u(a) \leq a + 1$ holds on $-1 \leq a \leq 1$, we have

$$|\mathcal{C}(x)| \leq K\hat{\eta} + K - \sum_{k=1}^{K} V(x,k) = K(\hat{\eta} - \bar{V}(\hat{\pi}(x)) + 1).$$

Transform the above equation into the expectation form, we get

$$\mathbb{E}[|\mathcal{C}(X)|] \leq K(\hat{\eta} - \mathbb{E}[\bar{V}(\hat{\pi}(X))] + 1).$$

Furthermore, for $\mathcal{C}_{\hat{\eta}} = \{(X,Y) \mid V(X,Y) \geq \hat{\eta}\}$ and its complementary set $\bar{\mathcal{C}}_{\hat{\eta}}$, we have

$$\mathbb{E}[\bar{V}(\hat{\pi}(X))] = (1-\alpha)\mathbb{E}[\bar{V}(\hat{\pi}(X)) \mid V(X,Y) \leq \hat{\eta}] + \alpha\mathbb{E}[\bar{V}(\hat{\pi}(X)) \mid V(X,Y) > \hat{\eta}],$$

and put it and the bound of $\hat{\eta}$ into the above equation:

$$\frac{1}{K}\mathbb{E}(|\mathcal{C}(X)|) \leq \mathbb{E}[\bar{V}(\hat{\pi}(X)) \mid \mathcal{C}_{\hat{\eta}}] - (1-\alpha)\mathbb{E}[\bar{V}(\hat{\pi}(X)) \mid \bar{\mathcal{C}}_{\hat{\eta}}] - \alpha\mathbb{E}[\bar{V}(\hat{\pi}(X)) \mid \mathcal{C}_{\hat{\eta}}] + C$$

$$= (1-\alpha)\big(\mathbb{E}[\bar{V}(\hat{\pi}(X)) \mid \mathcal{C}_{\hat{\eta}}] - \mathbb{E}[\bar{V}(\hat{\pi}(X)) \mid \bar{\mathcal{C}}_{\hat{\eta}}]\big) + C,$$

where constant $C := \mathbb{E}[\sqrt{2\big(H(\pi(X)) + \log(K)\big)} \mid \mathcal{C}_{\hat{\eta}}] + \tau + 1$. Using the lower bound $\bar{V}(\hat{\pi}(x)) \geq \frac{K+1}{2K}$, we have

$$\frac{1}{K}\mathbb{E}(|\mathcal{C}(X)|) \leq (1-\alpha)\big(\mathbb{E}[\bar{V}(\hat{\pi}(X)) \mid \mathcal{C}_{\hat{\eta}}] - \frac{K+1}{2K}\big) + C.$$

Given the assumption that $\mathbb{P}(H(\hat{\pi}(X)) \geq \frac{1}{2}C_K \mid \mathcal{C}_{\hat{\eta}}) \geq \mu$ ($\mu \in [0,1]$) and $\mathcal{D} := \{(X,Y) \mid H(\hat{\pi}(X)) \geq \frac{1}{2}C_K\}$, we obtain that

$$\frac{1}{K}\mathbb{E}(|\mathcal{C}(X)|) \leq (1-\alpha)\mathbb{E}[\bar{V}(\hat{\pi}(X)) \mid \mathcal{C}_{\hat{\eta}}] - (1-\alpha)\frac{K+1}{2K} + C$$

$$\leq (1-\alpha)\big(\mu(C_K - 1 - \mathbb{E}[H(\hat{\pi}(x)) \mid \mathcal{D} \cap \mathcal{C}_{\hat{\eta}}])$$

$$+ (1-\mu)(1 + \mathbb{E}[H(\hat{\pi}(x)) \mid \bar{\mathcal{D}} \cap \mathcal{C}_{\hat{\eta}}])\big) - (1-\alpha)\frac{K+1}{2K} + C$$

$$\leq (1-\alpha)(1-2\mu)\mathbb{E}[H(\hat{\pi}(x)) \mid \mathcal{C}_{\hat{\eta}}]$$

$$+ (1-\alpha)(\mu C_K - \frac{K+1}{2K}) + (1-\alpha)(1-2\mu) + C$$

By combining with the inequality derived in Proposition 1, where the final step uses $\mathbb{E}[H(\hat{\pi}(x)) \mid \bar{\mathcal{D}} \cap \mathcal{C}_{\hat{\eta}}] \leq \mathbb{E}[H(\hat{\pi}(x)) \mid \mathcal{C}_{\hat{\eta}}] \leq \mathbb{E}[H(\hat{\pi}(x)) \mid \mathcal{D} \cap \mathcal{C}_{\hat{\eta}}]$. $\qquad\square$

# B  FURTHER EXPERIMENT DETAILS

## B.1  DATASETS, MODELS, AND HYPERPARAMETERS

Table 4: Hyperparameters of adapters in conformal correction methods.

| DATASET | CIFAR10 | CIFAR100 | CORA-ML | CS | PHOTOS | TRUTHFULQA |
|---|---|---|---|---|---|---|
| MODEL | MLP | MLP | GAT | SGC | GAT | MLP |
| NUMBER OF LAYERS | 2 | 2 | 2 | 1 | 4 | 2 |
| HIDDEN DIMENSION | 128 | 256 | 64 | 32 | 16 | 128 |
| EPOCH | 200 | 500 | 5000 | 5000 | 5000 | 200 |
| BATCH SIZE | 512 | 1024 | - | - | - | - |
| LEARNING RATE | 0.0001 | 0.0001 | 0.0001 | 0.0001 | 0.001 | 0.001 |
| DROPOUT | - | - | 0.5 | 0.5 | 0.5 | - |
| WEIGHT DECAY | 1E-4 | 1E-4 | 5E-4 | 5E-4 | 5E-4 | 1E-4 |

We evaluate our method and baselines on five datasets across two domains: CIFAR10, CIFAR100, Cora-ML, CS, and Photos. The former two datasets are from the computer vision (CV) domain and the latter three are graph-structure datasets. For CV datasets, CIFAR10 and CIFAR100 consist of 60,000 $32 \times 32$ colour images in 10 and 100 classes, respectively. For graph datasets, there are 2,995/18,333/7,650 nodes with 2,879/6,805/745 features and 16,346/163,788/238,162 edges in Cora-ML, CS, and Photos. The number of classes in these graph datasets is 7, 15 and 8, respectively.

For CV tasks, we apply ResNet (He et al., 2016a), PreResNet (He et al., 2016b), and DenseNet (Huang et al., 2017) as the base models, and use MLP as the conformal adapter model. For graph tasks, we use GCN (Kipf & Welling, 2016), GAT (Velickovic et al., 2017), and SGC (Wu et al., 2019) as the base models and use GAT as the conformal adapter model following Huang et al. (2024b). For a fair comparison, we train each of the base models five times and report the average results to avoid fluctuations from randomness.

For base models, we strictly follow the settings in pytorch-classification[3] and CF-GNN (Smith, 2024) to pre-train CV models and graph models as base models, respectively. We also use open-source LLM Llama-2-7b-chat (Touvron et al., 2023) as the base model for the question answering task. The hyperparameters of adapters used in conformal correction methods on different datasets are listed in Table 4.

## B.2  EFFICIENCY COMPARSION WITH $\alpha = 0.2$

To study the influence of $\alpha$ on conformal correction, we report the results of marginal coverage and efficiency when $\alpha = 0.2$ on CV and graph datasets in Table 5. Generally speaking, the proposed EC[3] still outperforms other baselines on both CV and graph datasets. Specifically, EC[3] significantly improves the efficiency of APS by up to 52.5% on all datasets except CIFAR10. On CIFAR10, since the accuracy of base models is relatively high (e.g., $0.89$ in DenseNet100), conformal correction methods easily ameliorate the efficiency of both our method EC[3] and baseline ConfTr to achieve good efficiency.

## B.3  ENTROPY RESULTS OF CONFORMAL CORRECTION

The entropy thresholds are 3.03, 6.36, 2.52, 3.62, and 2.71 in CIFAR10, CIFAR100, Cora-ML, CS and Photos, respectively. Table 6 and Table 7 present the entropy result of Table 1, Table 2 and Table 5. Overall, EC[3] obtains a better balance between efficiency and entropy than other baselines with explicitly modeling the entropy of prediction results.

## B.4  ACCURACY RESULTS OF CONFORMAL CORRECTION

We list the average accuracy results of different algorithms on the test data in Table 8. We include ConfTr and CF-GNN as a reference on CV and graph datasets, respectively. The results illustrate

---

[3]The pytorch-classification repository is a popular Github project to implement the classification on CIFAR10/100; https://github.com/bearpaw/pytorch-classification.

Table 5: The efficiency results of conformal correction methods on CV and graph datasets when $\alpha = 0.2$. The results are the average of five runs of pre-trained model, each with 100 runs of conformal splits on datasets. Baselines are ConfTr and CF-GNN on CV and graph datasets, respectively. The best results are in shadow.

| DATASET | MODEL | CP | | BASELINE | | EC³ | |
|---|---|---|---|---|---|---|---|
| | | COVERAGE | EFFICIENCY | COVERAGE | EFFICIENCY | COVERAGE | EFFICIENCY |
| CIFAR10 | RESNET56 | $0.80_{\pm.00}$ | $4.01_{\pm.04}$ | $0.80_{\pm.00}$ | $1.06_{\pm.01}$ | $0.80_{\pm.00}$ | $1.04_{\pm.01}$ |
| | PRERESNET110 | $0.80_{\pm.00}$ | $4.22_{\pm.09}$ | $0.80_{\pm.00}$ | $1.01_{\pm.02}$ | $0.80_{\pm.00}$ | $1.02_{\pm.01}$ |
| | DENSENET100 | $0.80_{\pm.00}$ | $4.20_{\pm.02}$ | $0.80_{\pm.00}$ | $0.99_{\pm.00}$ | $0.80_{\pm.00}$ | $0.98_{\pm.00}$ |
| CIFAR100 | RESNET56 | $0.80_{\pm.00}$ | $13.38_{\pm.36}$ | $0.80_{\pm.00}$ | $6.17_{\pm.75}$ | $0.80_{\pm.00}$ | $2.93_{\pm.09}$ |
| | PRERESNET110 | $0.80_{\pm.00}$ | $14.98_{\pm.24}$ | $0.80_{\pm.00}$ | $4.72_{\pm.10}$ | $0.80_{\pm.00}$ | $4.25_{\pm.17}$ |
| | DENSENET100 | $0.80_{\pm.00}$ | $18.13_{\pm1.28}$ | $0.80_{\pm.00}$ | $3.20_{\pm.09}$ | $0.80_{\pm.00}$ | $2.76_{\pm.16}$ |
| CORA-ML | GCN | $0.80_{\pm.00}$ | $3.10_{\pm.23}$ | $0.80_{\pm.00}$ | $1.41_{\pm.10}$ | $0.80_{\pm.00}$ | $1.03_{\pm.03}$ |
| | GAT | $0.80_{\pm.00}$ | $3.07_{\pm.11}$ | $0.80_{\pm.00}$ | $1.47_{\pm.15}$ | $0.80_{\pm.00}$ | $1.10_{\pm.06}$ |
| | SGC | $0.80_{\pm.00}$ | $3.11_{\pm.15}$ | $0.80_{\pm.00}$ | $1.42_{\pm.12}$ | $0.80_{\pm.00}$ | $1.03_{\pm.04}$ |
| CS | GCN | $0.80_{\pm.00}$ | $6.12_{\pm.23}$ | $0.80_{\pm.00}$ | $2.66_{\pm.12}$ | $0.80_{\pm.00}$ | $1.73_{\pm.14}$ |
| | GAT | $0.80_{\pm.00}$ | $5.18_{\pm.19}$ | $0.80_{\pm.00}$ | $2.53_{\pm.23}$ | $0.80_{\pm.00}$ | $1.74_{\pm.17}$ |
| | SGC | $0.80_{\pm.00}$ | $6.15_{\pm.25}$ | $0.80_{\pm.00}$ | $2.41_{\pm.29}$ | $0.80_{\pm.00}$ | $1.84_{\pm.15}$ |
| PHOTOS | GCN | $0.80_{\pm.00}$ | $3.11_{\pm.14}$ | $0.80_{\pm.00}$ | $1.58_{\pm.31}$ | $0.80_{\pm.00}$ | $1.24_{\pm.07}$ |
| | GAT | $0.80_{\pm.00}$ | $1.83_{\pm.19}$ | $0.80_{\pm.00}$ | $1.74_{\pm.47}$ | $0.80_{\pm.00}$ | $1.61_{\pm.12}$ |
| | SGC | $0.80_{\pm.00}$ | $3.13_{\pm.05}$ | $0.80_{\pm.00}$ | $1.44_{\pm.28}$ | $0.80_{\pm.00}$ | $1.35_{\pm.09}$ |

Table 6: The entropy results of conformal correction methods on CV and graph datasets when $\alpha = 0.1$. The results are the average of five runs of pre-trained model, each with 100 runs of conformal splits on datasets. Baselines are ConfTr and CF-GNN on CV and graph datasets, respectively.

| DATASET | MODEL | CP | BASELINE | EC³ |
|---|---|---|---|---|
| CIFAR10 | RESNET56 | $0.19_{\pm.01}$ | $2.73_{\pm.40}$ | $2.97_{\pm.10}$ |
| | PRERESNET110 | $0.17_{\pm.01}$ | $2.79_{\pm.40}$ | $2.99_{\pm.19}$ |
| | DENSENET100 | $0.16_{\pm.00}$ | $2.55_{\pm.58}$ | $2.94_{\pm.50}$ |
| CIFAR100 | RESNET56 | $0.62_{\pm.09}$ | $2.93_{\pm.51}$ | $4.26_{\pm.61}$ |
| | PRERESNET110 | $0.67_{\pm.03}$ | $3.16_{\pm1.06}$ | $4.52_{\pm.48}$ |
| | DENSENET100 | $0.57_{\pm.02}$ | $2.49_{\pm.40}$ | $4.14_{\pm.48}$ |
| CORA-ML | GCN | $0.74_{\pm.28}$ | $2.12_{\pm.37}$ | $2.32_{\pm.01}$ |
| | GAT | $0.76_{\pm.11}$ | $2.29_{\pm.30}$ | $2.37_{\pm.02}$ |
| | SGC | $0.76_{\pm.14}$ | $2.30_{\pm.16}$ | $2.33_{\pm.02}$ |
| CS | GCN | $0.40_{\pm.07}$ | $3.40_{\pm.10}$ | $3.00_{\pm.22}$ |
| | GAT | $0.57_{\pm.11}$ | $3.39_{\pm.11}$ | $2.90_{\pm.27}$ |
| | SGC | $0.49_{\pm.07}$ | $3.39_{\pm.08}$ | $2.93_{\pm.32}$ |
| PHOTOS | GCN | $0.62_{\pm.04}$ | $2.28_{\pm.76}$ | $2.19_{\pm.03}$ |
| | GAT | $1.32_{\pm.18}$ | $2.33_{\pm.73}$ | $2.25_{\pm.04}$ |
| | SGC | $0.66_{\pm.04}$ | $2.22_{\pm.84}$ | $2.13_{\pm.03}$ |

that: (1) the accuracy decreases for all conformal correction algorithms; (2) our proposed method EC³ is generally comparable to the conformal correction baselines on both CV and graph domains, and achieves a better balance between efficiency and entropy.

Table 7: The entropy results of conformal correction methods on CV and graph datasets when $\alpha = 0.2$. The results are the average of five runs of pre-trained model, each with 100 runs of conformal splits on datasets. Baselines are ConfTr and CF-GNN on CV and graph datasets, respectively.

| DATASET | MODEL | CP | BASELINE | EC$^3$ |
|---|---|---|---|---|
| | RESNET56 | $0.19\pm.01$ | $2.99\pm.00$ | $2.98\pm.01$ |
| CIFAR10 | PRERESNET110 | $0.17\pm.01$ | $2.94\pm.03$ | $2.99\pm.01$ |
| | DENSENET100 | $0.16\pm.00$ | $2.97\pm.02$ | $2.99\pm.01$ |
| | RESNET56 | $0.64\pm.09$ | $1.73\pm.11$ | $2.93\pm.09$ |
| CIFAR100 | PRERESNET110 | $0.65\pm.03$ | $1.60\pm.05$ | $2.64\pm.07$ |
| | DENSENET100 | $0.56\pm.01$ | $1.24\pm.04$ | $2.56\pm.19$ |
| | GCN | $0.74\pm.28$ | $2.34\pm.06$ | $2.39\pm.18$ |
| CORA-ML | GAT | $0.76\pm.11$ | $2.39\pm.03$ | $2.38\pm.13$ |
| | SGC | $0.76\pm.14$ | $2.35\pm.05$ | $2.46\pm.20$ |
| | GCN | $0.40\pm.07$ | $3.37\pm.02$ | $3.56\pm.08$ |
| CS | GAT | $0.57\pm.11$ | $3.38\pm.02$ | $3.56\pm.14$ |
| | SGC | $0.49\pm.07$ | $3.39\pm.01$ | $3.58\pm.07$ |
| | GCN | $0.62\pm.04$ | $2.23\pm.75$ | $2.20\pm.02$ |
| PHOTOS | GAT | $1.32\pm.18$ | $2.70\pm.46$ | $2.25\pm.03$ |
| | SGC | $0.66\pm.04$ | $2.18\pm.84$ | $2.12\pm.02$ |

Table 8: The accuracy results of conformal correction methods on CV and graph datasets.

| DATASET | MODEL | CP | BASELINE | EC$^3$ |
|---|---|---|---|---|
| | RESNET56 | 84.90 | 84.72 | 84.76 |
| CIFAR10 | PRERESNET110 | 86.07 | 85.93 | 85.82 |
| | DENSENET100 | 88.63 | 88.41 | 88.09 |
| | RESNET56 | 66.45 | 62.84 | 62.28 |
| CIFAR100 | PRERESNET110 | 69.11 | 64.58 | 65.52 |
| | DENSENET100 | 74.10 | 67.68 | 67.92 |
| | GCN | 88.55 | 85.74 | 83.51 |
| CORA-ML | GAT | 85.57 | 83.92 | 81.15 |
| | SGC | 87.20 | 86.00 | 83.28 |
| | GCN | 94.28 | 77.05 | 89.92 |
| CS | GAT | 92.74 | 78.04 | 89.58 |
| | SGC | 93.41 | 78.42 | 88.62 |
| | GCN | 93.26 | 77.70 | 82.84 |
| PHOTOS | GAT | 92.32 | 67.75 | 69.17 |
| | SGC | 92.82 | 72.96 | 83.61 |

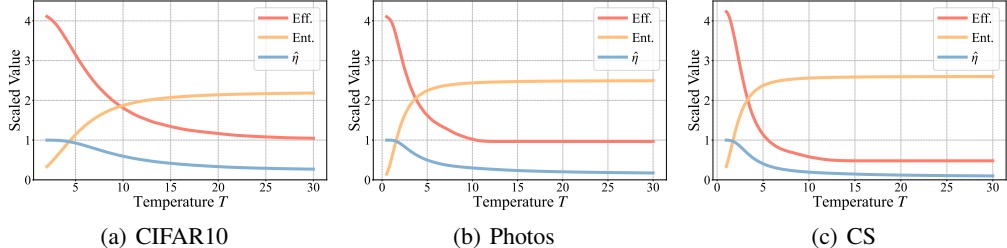

(a) CIFAR10     (b) Photos     (c) CS

Figure 5: Efficiency and entropy results of APS on the test set after temperature scaling w.r.t. $T$ when $\alpha = 0.1$.

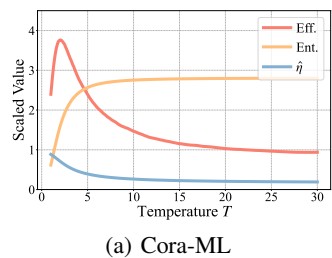
(a) Cora-ML

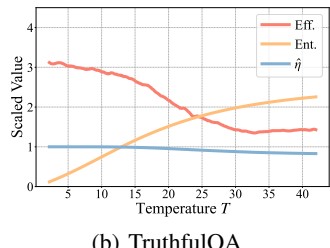
(b) TruthfulQA

Figure 6: Efficiency and entropy results of RAPS (left) and APS (right) on the test set after temperature scaling w.r.t. $T$.

Table 9: The coverage, entropy and efficiency results of conformal correction methods on CIFAR10 and Cora-ML with RAPS when $\alpha = 0.1$. The results are the average of five runs of pre-trained model, each with 100 runs of conformal splits on both datasets. Baselines are ConfTr and CF-GNN on CIFAR10 and Cora-ML, respectively.

| METRIC | DATASET | CP | BASELINE | EC$^3$ |
|---|---|---|---|---|
| COVERAGE | CIFAR10 | $0.90_{\pm.00}$ | $0.90_{\pm.00}$ | $0.90_{\pm.00}$ |
| | CORA-ML | $0.90_{\pm.00}$ | $0.90_{\pm.00}$ | $0.90_{\pm.00}$ |
| EFFICIENCY | CIFAR10 | $1.47_{\pm.01}$ | $1.41_{\pm.03}$ | $1.29_{\pm.02}$ |
| | CORA-ML | $1.54_{\pm.12}$ | $1.44_{\pm.04}$ | $1.38_{\pm.02}$ |
| ENTROPY | CIFAR10 | $0.19_{\pm.01}$ | $2.90_{\pm.04}$ | $3.02_{\pm.05}$ |
| | CORA-ML | $1.04_{\pm.31}$ | $0.60_{\pm.01}$ | $1.60_{\pm.02}$ |

## B.5 ADDITIONAL RESULTS OF TEMPERATURE SCALING

In this part, we offer the efficiency and entropy results for the remaining three datasets with temperature $T$ growing, i.e., CIFAR10, CS, and Photos. Similar to Fig. 2(b) and 2(c), as $T$ rises, temperature scaling improves the efficiency of APS and covers almost the entire value range of entropy, which demonstrates its strong control over entropy.

## B.6 EFFICIENCY COMPARSION WITH REGULARIZED ADAPTIVE PREDICTION SETS

To further demonstrate our methods' adaptability to various adaptive conformal prediciton approaches, we conduct experiments using RAPS, which regularizes APS to generate a smaller prediction set size. The performance results when $\alpha = 0.1$ are reported in Table 9. We find that the proposed EC$^3$ outperforms the baselines on both CIFAR10 and Cora-ML. In addition, the results of RAPS after temperature scaling further confirm our Theorem 3, as shown in Fig. 6(a).

## B.7 CONDITIONAL COVERAGE OF CONFORMAL CORRECTION

Here, we present the conditional coverage results before and after conformal correction by the metrics WSC and SSCV in Table 10. It is clear that conformal correction can tacitly improve conditional coverage to some extent.

## B.8 PARAMETER SENSITIVITY

We next perform parameter sensitivity analysis on the two entropy-controlled hyperparameters of our model, i.e., $T$, the temperature, and $\gamma$, which controls the importance of the entropy term. For $T$, we pick sufficient points to cover the entropy range and let $\gamma \in \{2, 4, 6, 8, 10\}$. The results are shown in Fig. 7, where we plot TS, EC$^3$-1, and EC$^3$-2 on CIFAR10 and Cora-ML for brevity. We observe that there exists a trade-off between efficiency and entropy when the model entropy changes with either $T$

Table 10: Conditional coverage results on CIFAR10 and Cora-ML before/after conformal correction. WSC closer to 0.9 and smaller SSCV indicate better performance. Conformal correction does not worsen the conditional coverage.

| DATASET | METHOD | WSC | SSCV |
|---|---|---|---|
| CIFAR10 | CP | $0.88_{\pm.01}$ | $0.45_{\pm.01}$ |
| | CONFTR | $0.89_{\pm.00}$ | $0.22_{\pm.18}$ |
| | TS | $0.88_{\pm.01}$ | $0.19_{\pm.02}$ |
| | $EC^3$ | $0.89_{\pm.00}$ | $0.15_{\pm.05}$ |
| CORA-ML | CP | $0.90_{\pm.00}$ | $0.10_{\pm.00}$ |
| | CF-GNN | $0.90_{\pm.00}$ | $0.09_{\pm.05}$ |
| | TS | $0.90_{\pm.00}$ | $0.09_{\pm.01}$ |
| | $EC^3$ | $0.90_{\pm.00}$ | $0.09_{\pm.05}$ |

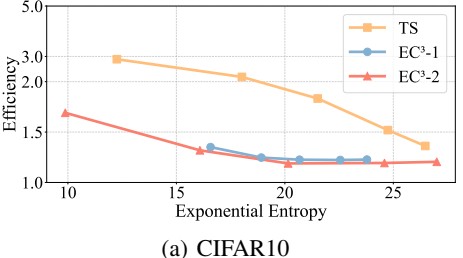
(a) CIFAR10

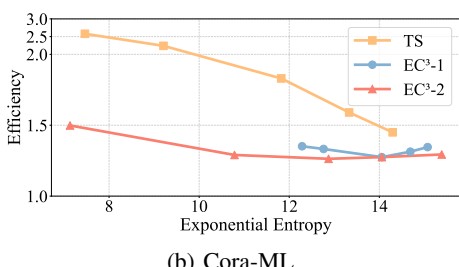
(b) Cora-ML

Figure 7: Parameter sensitivity analysis. Temperature Scaling (TS) and $EC^3$-2 (with TS) both change with temperature $T$, while $EC^3$-1 (without TS) varies with hyperparameter $\gamma$.

or $\gamma$. Additionally, the Pareto frontier of $EC^3$-2 is significantly better than that of TS in both Fig. 7(a) and 7(b), which indicates the importance of conformal correction networks in our method.

## B.9 EVALUATION ON LLMS

We evaluate our approach on the question answering task using the TruthfulQA dataset (Lin et al., 2022). The prompt we use is shown as follows.

> This is a 4-choice question that you should answer:{question}. Put the final results within \boxed{{}}, e.g., \boxed{{A}}. The correct answer to this question is:".

For each question, we sample 100 Chain-of-Thought responses from Llama-2-7b-chat (Touvron et al., 2023) and record the answer distribution (based on self-consistency) to perform conformal prediction. Table 11 shows the results of APS when $\alpha = 0.2$, considering the low accuracy 48.2%, and our method obtains better efficiency than the baseline ConfTr at the same level of entropy. Moreover, the results after temperature scaling are provided in Fig. 6(b). The experimental results demonstrate that the efficiency-entropy trade-off of conformal prediction is also present in LLM generation tasks.

Table 11: The coverage, entropy and efficiency results of conformal correction methods on LLMs for the question answering task.

| METRIC | CP | CONFTR | EC$^3$ |
|---|---|---|---|
| COVERAGE | $0.81_{\pm.00}$ | $0.80_{\pm.00}$ | $0.81_{\pm.00}$ |
| EFFICIENCY | $2.92_{\pm.01}$ | $2.48_{\pm.08}$ | $2.32_{\pm.15}$ |
| ENTROPY | $0.04_{\pm.01}$ | $1.84_{\pm.11}$ | $1.89_{\pm.07}$ |

