# OpenReview forum: "Conformal Correction for Efficiency May be at Odds with Entropy"
_ICLR.cc/2026/Conference — Submitted to ICLR 2026_

### Official Review · Reviewer_wJCq · 2025-10-26

**Soundness:** 3
**Presentation:** 3
**Contribution:** 2
**Rating:** 4
**Confidence:** 4

**Summary:**

The paper shows a fundamental trade-off in conformal prediction between efficiency (smaller sets) and prediction entropy (more decisive probabilities), and reframes conformal correction as multi-objective optimization on this Pareto frontier. It introduces EC3, a plug-in adapter that takes base probabilities and is trained with a conformal-aware inefficiency loss while constraining entropy via focal loss and temperature scaling, shrinking sets without inflating uncertainty. A theoretical link between APS expected set size and entropy explains why improving one typically harms the other. Experiments on vision and graph benchmarks indicate EC3 consistently yields better efficiency–entropy trade-offs and improves conditional coverage while preserving target marginal coverage.

**Strengths:**

1. This paper clearly identifies and formalizes a trade-off between CP efficiency and prediction entropy, with supporting analysis (e.g., APS-based bounds) that explains why the objectives can conflict.
2. This paper persuasively recasts conformal correction as a multi-objective problem with entropy as a first-class constraint, making the practical value obvious.
3. The paper is well organized; figures (especially Pareto plots) effectively illustrate key ideas and make the theory easy to follow.

**Weaknesses:**

1. The contribution is hard to assess because similar ideas have been explored previously. Please expand the related-work discussion to delineate what is new here and add the relevant prior methods as baselines to the empirical evaluation.
2. This work does not establish formal conformal coverage guarantees, in contrast to standard CP.

[1] Xi H, Huang J, Liu K, et al. Does confidence calibration improve conformal prediction.
[2] Dabah L, Tirer T. On Temperature Scaling and Conformal Prediction of Deep Classifiers.

**Questions:**

1. See weaknesses. Please discuss more about related works and experimental baselines to outline the contribution.
2. Please provide more training details: loss trajectories and how the two objectives (efficiency vs. entropy) evolve and interact during optimization.
3. The evaluation shows that, at matched entropy, your method yields smaller prediction sets. Why is this desirable for downstream decision making?
4. Hyperparameters appear important; please expand discussion and provide practical selection guidelines.
5. How does the method perform under distribution shift?

---

> ### Author Response · Authors · 2025-12-02
>
> We thank Reviewer wJCq for their insightful feedback and address your concerns below.
>
> > **On Weakness 1 & Question 1 (discuss more about related works and experimental baselines)**
>
> For ConfTS [1], we have discussed this work in the related work (see Line 463). To empirically validate our method, we also conducted a comparative experiment for temperature scaling, analogous to ConfTS, and $EC^{3}$ in Figure 7. Our $EC^{3}$ (red line) achieved a better Pareto frontier of efficiency and entropy than ConfTS (yellow line).
>
> Regarding the paper of Dabah and Tirer [2], we thank the reviewer for pointing out the valuable reference. To clarify, Dabah and Tirer discuss the relationship between the model calibration (i.e., temperature scaling) and conformal prediction, as similar as [1]. Different from [1,2], we find that a better balance between efficiency and entropy can be achieved via controlling entropy during conformal correction. Furthermore, we also provide a comprehensive theoretical framework to analyze the essential trade-off between efficiency and entropy, not limited to temperature scaling.
>
> > **On Weakness 2 (establishing formal conformal coverage guarantees)**
>
> We would like to clarify that conformal correction is a preprocessing step for conformal prediction, having no impact on the original guarantee of conformal prediction. This is also confirmed by our experimental results in Table 1 and Table 2.
>
>
> > **On Question 2 (more training details)**
>
> As the epoch increases, the training loss of $EC^{3}$ initially drops, then stabilizes. Additionally, the trajectory of two objectives, efficiency and entropy, is as similar as that in Figure 1(b) of our paper.
>
>
> > **On Question 3 (low entropy is desirable)**
>
> Please see Q2 in General Response.
>
> > **On Question 4 (hyperparameter selection)**
>
> The parameters $\gamma$ and $T$ govern the trade-off between efficiency and entropy, which can be tuned based on application needs. For example, increasing $\gamma$ and $T$ prioritizes efficiency at the cost of more entropy, while decreasing them favors lower entropy at the expense of efficiency.
>
> > **On Question 5 (distribution shift)**
>
> We would like to clarify that this work is based on the exchangeability assumption, which is implicit in conformal prediction [3]. We plan to extend our work to move beyond exchangeability in the future work.
>
> ---
> ### References
> [1] Does confidence calibration improve conformal prediction? Xi et. al, TMLR 2025
>
> [2] On Temperature Scaling and Conformal Prediction of Deep Classifiers. Dabah et. al, ICML 2025
>
> [3] Algorithmic learning in a random world. Vovk et. al, Springer 2005

---

### Official Review · Reviewer_s5ab · 2025-10-27

**Soundness:** 2
**Presentation:** 3
**Contribution:** 2
**Rating:** 4
**Confidence:** 4

**Summary:**

The paper introduces Entropy-Constrained Conformal Correction (EC$^3$), an approach to improve the efficiency of non-conformity score “adapters” for downstream conformal prediction. The authors facilitate this by augmenting the inefficiency loss used to train these adapters with focal loss. Lastly, they provide theoretical grounding for adding focal loss and empirical results using EC$^3$ on image classification, node classification, and Q&A tasks.

**Strengths:**

- The paper includes a strong empirical evaluation, presenting results across several datasets with varying characteristics (i.e., number of classes).
- The paper provides a clear justification for using **focal loss** in training adapters for Adaptive Prediction Sets (APS) non-conformity scores. It also provides a good theoretical connection between minimizing focal loss and maximizing entropy, leading to smaller prediction set sizes (**Theorem 3**, under the assumption $\mu \geq 0.5$).
- The paper also provides a **class-conditional version of EC³** to address imbalanced class coverage.

**Weaknesses:**

- The main weakness of the paper is the lack of comparison with APS using randomization, as introduced in [1]. The authors employ APS **without randomization** in their experiments. In other words, their implementation omits the second red and bolded term in the randomized non-conformity score:
$$
V(x, y; u) = \sum_{i=1}^{y} \hat{\pi}_ {(i)}(x)~\textcolor{red}{\mathbf{- u \hat{\pi}_{(y)}(x) }}
$$
for
$u \sim U([0,1])$. Theoretically and empirically, randomized APS has been shown to **reduce set size**  (**improve efficiency**) compared to non-randomized APS [2]; thus, it is imperative to include randomized APS when comparing with SOTA methods.

   - For instance, Cora-ML is reported to have efficiencies of around 4 and 1.85 for the baseline CP and CF-GNN methods in Table 2, respectively, using non-randomized APS. However, Figure 5 in [2] shows efficiencies closer to 1.5 when using the randomized APS approach for both baseline CP and CF-GNN — indicating that randomized APS performs similarly to EC³. (Similar comparisons are available in [2] for the remaining datasets). For this reason, the perceived gains of **EC³** may not be as significant as claimed.

- The paper is missing an efficiency plot/table for the class-conditional $EC^3$. It is important to quantify the efficiency and SSCV trade-off between the two methods.

- The paper references $EC^{3}-1$ and $EC^{3}-2$, but it is not clear what those are referencing.

- Missing definitions for SSCV and WSC, either in the main body or appendix.


References

[1] Y. Romano et al. Classification with valid and adaptive coverage [NeurIPS 202]

[2] P. Maneriker et al. Conformal Prediction: A Theoretical Note and Benchmarking Transductive Node Classification in Graphs [TMLR 2025]

**Questions:**

See weaknesses. The main weakness/question is a lack of comparison with APS with randomization. If that can be addressed by the authors and the $EC^3$ still provides quality results, I will be happy to adjust my score accordingly.

L743: "inequation" -> "inequality".

---

> ### Author Response · Authors · 2025-12-02
>
> We sincerely thank Reviewer s5ab for their insightful comments. We address your concerns below.
>
> > **On Weakness 1 & Question 1 (APS using randomization)**
>
> Thank you for this valuable comment. Following your suggestion, we conduct a comparative experiment on randomized APS with temperature scaling to control entropy, and list the efficiency results when their entropy is comparable in the table. The results demonstrate that our $EC^{3}$ still achieves a better balance between efficiency and entropy than CFGNN on randomized APS. We will include this result into the revised version.
>
> |Method   |CP |CFGNN |EC3 |
> |:------|:---:|:---:|:---:|
> |Efficiency|1.60|1.50|1.45|
> |Entropy|1.33|1.33|1.24|
>
>
> > **On Weakness 2 (quantify the efficiency and SSCV trade-off)**
>
> We reported the SSCV results in Table 10. The experiments demonstrated that our $EC^{3}$ method not only maintains conditional coverage but even improves it on the CIFAR-10 dataset. To further assess class-conditional coverage, we employed the WSC metric, which identifies the subgroup (or "slab") with the worst coverage. As shown in Table 10, $EC^{3}$ also preserves performance under this stricter metric, indicating almost no negative effect on the worst-slab coverage.
>
>
> > **On Weakness 3 ($EC^{3}-1$ and $EC^{3}-2$)**
>
> Thanks for the comment. $EC^{3}-2$ (with TS) and $EC^{3}-1$ (without TS) respectively control entropy by different parameters: the former by temperature $T$, and the latter by hyperparameter $\gamma$ as mentioned in the caption of Figure 7. We will refine this description to be more precise in the revised version.
>
>
> > **On Weakness 4 (missing definitions for SSCV and WSC)**
>
> SSCV [1] and WSC [2] are two common conditional coverage metrics. SSCV evaluates coverage across prediction sets of different sizes, while WSC identifies the subgroup with the worst coverage through a grid search. References for both metrics were provided in Line 354 of the paper.
>
> -----
> ### References
> [1] Uncertainty Sets for Image Classifiers using Conformal Prediction. Angelopoulos et. al, ICLR 2021
>
> [2] Classification with valid and adaptive coverage. Romano et. al, NeurIPS 2020

---

### Official Review · Reviewer_G9pT · 2025-11-01

**Soundness:** 3
**Presentation:** 3
**Contribution:** 2
**Rating:** 4
**Confidence:** 3

**Summary:**

This paper investigates the relationship between efficiency (small size of conformal prediction (CP) sets) and entropy (uncertainty in model predictions) in conformal prediction frameworks. The authors show empirically and theoretically that these two quantities are often in conflict: increasing CP efficiency typically raises prediction entropy, reducing confidence in predictions.

To address this, the authors introduce $EC^3$ (Entropy-Constrained Conformal Correction), a new conformal correction method that adds an entropy control term (via focal loss and temperature scaling) to balance efficiency and entropy. They formalize this trade-off as a Pareto frontier and use $EC^3$ to search for better optima.

Experiments on vision (CIFAR-10/100) and graph datasets (Cora-ML, CS, Photos) show that $EC^3$ achieves up to 34% improvement in efficiency while maintaining marginal coverage and improving conditional coverage.

**Strengths:**

- The paper provides a first rigorous analysis linking CP inefficiency and prediction entropy (Propositions 1-2, Theorem 3).
- It identifies a real tension between compact prediction sets and calibrated uncertainty, which has been largely ignored by prior conformal training work.
- The $EC^3$ objective combines focal loss and inefficiency regularization with entropy control; temperature scaling provides a simple yet effective Pareto traversal mechanism.
- Extensive experiments across multiple architectures and domains; results are consistent and show significant practical gains.
- The paper is clearly written, with good visualizations (e.g., Pareto frontiers) and an accessible discussion of theoretical results.

**Weaknesses:**

- The analysis focuses on adaptive conformal prediction (APS); extension to other CP variants (e.g., regression or non-adaptive scores) is not discussed.
- The entropy parameter $\gamma$ and the temperature $T$ are hyperparameters tuned via grid search; no principled guidance for choosing them is provided.
- While acknowledged in the Limitations section, empirical degradation in base model accuracy is not quantified or analyzed.
- Some proofs (especially Proposition 2 and Theorem 3) rely on simplifying assumptions (exchangeability, bounded calibration errors) that might not hold in practical non-i.i.d. data settings.
- Baselines are limited to conformal-training-based methods; recent calibration or information-theoretic conformal approaches could strengthen evaluation.

**Questions:**

- How sensitive is $EC^3$ to the choice of $\gamma$ (entropy weight) and $\beta$ (inefficiency weight)? Can adaptive schedules mitigate tuning difficulty?
- Does $EC^3$ preserve or degrade conditional coverage guarantees beyond empirical results? Are there any theoretical bounds?
- How does the method perform on non-classification tasks (e.g., regression CP or language model calibration)?
- Could temperature scaling alone (without $EC^3$) achieve comparable Pareto improvements with careful tuning?
- Are there insights into how entropy affects human interpretability of conformal sets - e.g., does lower entropy align with better human decision support?

---

> ### Author Response · Authors · 2025-12-02
>
> We thank Reviewer G9pT for the insightful feedback. Here are our responses to your concerns and questions.
>
> > **On Weakness 1 (other CP variants)**
>
> Please see Q1 in General Response.
>
>
> > **On Weakness 2 (the guidance for choosing $\gamma$ and $T$)**
>
> The parameters $\gamma$ and $T$ are used to balance the trade-off between efficiency and entropy, depending on the real application's requirements. For example, prioritizing efficiency over entropy is achieved by increasing $\gamma$ and $T$, which improves efficiency at the expense of more entropy, and vice versa.
>
>
> > **On Weakness 3 (accuracy results in base model accuracy)**
>
> We reported the accuracy results and analyzed the experimental results in Appendix B.4. Please see the corresponding paragraph for more details.
>
>
> > **On Weakness 4 (non-i.i.d. data settings)**
>
> We would like to clarify that this work concentrates on the exchangeability assumption, which is common in conformal prediction [3]. It is important to note that while the i.i.d. condition implies exchangeability, exchangeability itself is a weaker and more broadly applicable assumption, which covers most situations in machine learning.
>
>
> > **On Weakness 5 (limited to conformal-training-based methods)**
>
> We would like to clarify that in this work, we focus on the trade-off in conformal correction technique, characterized by using conformal training to improve efficiency, as introduced in abstraction. We leave the exploration of other conformal methods for future work.
>
>
> > **On Question 1 (the sensitivity of $\gamma$ and $\beta$)**
>
> For $\gamma$, we conducted the parameter sensitivity experiments for it in Appendix B.8. For $\beta$, it is fixed by 0.1 by default following existing work [1].
>
>
>
> > **On Question 2 (conditional coverage guarantees beyond empirical results)**
>
> In this work, we mainly focus on the trade-off between efficiency and entropy. The guarantee of conditional coverage is another important topic, and we leave it to future work.
>
>
> > **On Question 3 (non-classification tasks)**
>
> Please see Q1 in General Response.
>
>
> > **On Question 4 (temperature scaling alone)**
>
> We have conducted the ablation study to compare the temperature scaling and our $EC^{3}$ method in Figure 7. The results demonstrate that $EC^{3}$ achieves a better Pareto frontier than using temperature scaling alone.
>
>
> > **On Question 5 (insights into the effects of entropy)**
>
> Please see Q2 in General Response.
>
>
> -------
> ### References
> [1] Uncertainty Quantification over Graph with Conformalized Graph Neural Networks. Huang et. al, NeurIPS 2023

---

### Official Review · Reviewer_3wJp · 2025-11-05

**Soundness:** 2
**Presentation:** 2
**Contribution:** 2
**Rating:** 4
**Confidence:** 2

**Summary:**

This paper empirically highlights a tradeoff between the entropy of the model predictions and the set size ( efficiency ) of CP. The
 authors provide theoretical justifications of this trade-off for a specific score function and propose a new conformal training algorithms that achieves a more favorable balance between entropy and efficiency. The method, which fine-tunes pertained predictors using the proposed conformal adaptor, rather than, scratch, demonstrates improved efficiency in terms of prediction set size compared to existing conformal training methods such as ConfTr ( Stutz et al. )

**Strengths:**

1. The investigation of the entropy and CP set size in this particular setting of conformal training is novel to the best of my knowledge. the framing is new and can be valuable for future research.

2. The theoretical results, though specific to a single score function, are well-motivated and provide interesting intuition.

3. The empirical results show promise in terms of prediction set size minimization relative to prior work.

4. The authors had valuable practical considerations in mind. Instead of retraining the entire model, the paper utilizes the idea of conformal wrappers to finetune, which reduces the computational cost and improves practicality. this design choice suggests that the authors are mindful of real-world applicability.

**Weaknesses:**

1. **Limited scope of score functions**: the authors focus exclusively on the APS-type score functions. it remains unclear whether the observed entropy-efficiency tradeoff generalizes to other commonly used conformal scores such as 1-p(y|x), which is standard in split-conformal methods. At minimum, empirical evidence across multiple score functions would significantly strengthen the claims.

2. Theoretical results are again presented only considering the APS score. While generalizing the analysis to other score functions may be challenging, it would be valuable to at least empirically test whether similar behaviors are observed with alternatives.

3. I found the evidence insufficient for the observed claimed trade-off. More explanations are needed. Particularly the explanation around lines 070-090 and Figure 1 does not convincingly establish the existence of the proposed tradeoff. It would be important to (i) evaluate whether the same trend holds under full model retraining, (ii) demonstrate the empirical tradeoff using other score functions. Moreover in this paragraph I found the explanation confusing for figure 1. does this figure correspond with only fine-tuning using L_class ? if only utilizing the L_class, then there is no term balancing conformal set size and thus its just standard fine-tuning using cross entropy loss ? I would appreciate clarification from the authors.

4. the alignment between text and figure is sometimes unclear, making it difficult to interpret how empirical results support the theoretical claims.

**Questions:**

1. **Line 158**: the authors mentions jointly optimizing L_class and L_efficiency. The figure only corresponds to L_class. ( which is standard cross entropy loss). It is possible to train ( even from scratch) using only L_efficiency to obtain smaller sets at comparable accuracy ( thats what conformal training does.) Can you show the results and figure when only using L_efficiency ?


2.  Figure 1a vs Line 160: In figure 1a, efficiency initially decreases while accuracy increases, yet the text suggests both increase together. Please clarify this discrepancy or update the figure/text for consistency.

3. Line 071 (CIFAR-100 results): It is counterintuitive that prediction set size increases after applying conformal correction. Could you clarify this behavior?

---

> ### Author Response · Authors · 2025-12-02
>
> We thank Reviewer 3wJp for the constructive feedback. We address your concerns as follows.
>
> > **On Weakness 1 & Weakness 2 (Limited scope of score functions)**
>
> Please see Q1 in General Response.
>
>
> > **On Weakness 3 (the insufficient evidence for the claimed trade-off)**
>
> We would like to clarify that the core of conformal correction [1] is to train a conformal-aware adapter on a fixed base model. This approach is more akin to traditional conformal prediction, as it remains decoupled from the base model's training, thereby offering broader practical applicability. Regarding the trade-offs for other score functions, we also evaluated RAPS and have reported its results in Appendix B.6. Please refer to the relevant section for further details.
>
> Furthermore, given that the standard conformal correction, using both L_class and L_inefficiency, constitutes a multi-objective optimization problem, it is natural to investigate whether these two objectives can achieve their minima simultaneously. To explore this, we conducted an experiment using only L_class (without L_inefficiency) and plotted the trajectories of accuracy, efficiency, and entropy in Figure 1(b). The results reveal that during the initial training stage, as conformal prediction benefits from the base model's good classification performance, both accuracy and efficiency improve while entropy decreases. However, once accuracy converges and the model's entropy drops further—indicating increased certainty of models—the efficiency begins to decrease. This observed phenomenon suggests a potential trade-off between efficiency (L_inefficiency) and entropy (L_class) in fully trained models.
>
>
> > **On Question 1 (training using only L_efficiency)**
>
> As suggested, we attempted to perform conformal correction using only the L_inefficiency loss. However, the models failed to converge due to extremely poor classification performance, leading to a meaningless conformal prediction. This result confirms that the L_class loss is indispensable for successful conformal correction.
>
>
> > **On Question 2 (Figure 1a vs Line 160)**
>
> We believe you are referring to Figure 1(b). The metric for efficiency is the average set size, where a smaller size indicates higher efficiency. Therefore, the initial training stage in Figure 1(b) shows a simultaneous improvement in both accuracy and efficiency, which aligns with the description in the text.
>
>
> > **On Question 3 (CIFAR-100 results in Line 071)**
>
> We would like to clarify that this result precisely reveals the trade-off we find between efficiency and entropy. We have confirmed it both theoretically and empirically. Consequently, conformal prediction typically employs an early stopping strategy to ensure compact prediction sets when entropy is high.
>
> -----
> ### References
> [1] Uncertainty Quantification over Graph with Conformalized Graph Neural Networks. Huang et. al, NeurIPS 2023

---

### Author Response · Authors · 2025-12-02
**General Response**

We sincerely thank all the reviewers for their valuable insights and constructive suggestions, which have greatly helped us strengthen the paper. We appreciate the reviewers’ recognition of our work, such as our valuable findings—efficiency-entropy trade-off (Reviewer 3wJp, G9pT, wJCq), the significance of our theoretical analysis (Reviewer 3wJp, G9pT, s5ab, wJCq), and the strong empirical evaluation (Reviewer 3wJp, G9pT, s5ab). We also appreciate Reviewer s5ab’s comments that he/she will be happy to adjust the score given the comparison with randomized APS. Below, we provide a summary of the key questions raised by the reviewers as well as our responses/revisions made to address them.

> **Scope (Reviewer 3wJp, G9pT).** Reviewers have questioned the limited scope of score functions and evaluation tasks.

Indeed, this is a limitation we admitted in the paper. Currently, our theoretical analysis only covers APS (which is still the mainstream choice for conformal classification tasks, taking both efficiency and conditional validity into account, and numerous non-conformity scores are defined in the APS family [1,2,3]), and extending it to the entire APS family is extremely difficult. However, empirically, we have extended to other scores in the APS family. For example, the experimental results of RAPS [2] were given in Appendix B.6 in our paper.  For SAPS [3], we take the temperature scaling as an example to conduct experiments and observe similar results as APS. The detailed results are shown in the [link](https://figshare.com/s/389a7833771fdac8d843). Additionally, since entropy is hard to define on regression tasks, we alternatively extend our experiments to generation tasks of LLMs on the TruthfulQA dataset. The results were given in Appendix B.9.

Still, we would like to emphasize the key contributions of this work. That is, we recognize a new dimension—entropy—into the field of conformal prediction, empirically and theoretically identify a trade-off between the CP efficiency and the entropy of model prediction (under the classic APS), and propose an entropy-constrained conformal correction method to explore a better Pareto optimum between efficiency and entropy.

> **Impact (Reviewer G9pT, wJCq).** Reviewers also questioned the significance of the entropy-efficiency tradeoff in decision making.

The key significance of our method is to provide a better entropy-efficiency frontier for classification tasks. Specifically, 1) efficiency (the size of prediction sets) measures how many choices we need to consider; 2) entropy measures how certain we are about the top-choices in the provided prediction set.

Consider the following example where a patient is presented with a fever symptom, and we obtain two prediction sets for the causes of the fever: *A: {Influenza, Rheumatism}* with predictive probabilities 0.4 and 0.4, and *B: {Influenza, Rheumatism, Pneumonia}* with predictive probabilities 0.6, 0.1, and 0.1, via balancing efficiency and entropy. If the true diagnosis is *Influenza*, set *B* provides a clearer diagnostic signal by assigning *Influenza* the highest confidence (0.6). This is particularly advantageous in clinical settings, where diagnostic trial-and-error carries high risks and costs.

-------
### References
[1] Classification with valid and adaptive coverage. Romano et. al, NeurIPS 2020

[2] Uncertainty sets for image classifiers using conformal prediction. Angelopoulos et. al, ICLR 2021

[3] Conformal prediction for deep classifier via label ranking. Huang et. al, ICML 2024

---

### Meta-Review · Area_Chair_f9xR · 2026-01-10

**Summary:**

This paper considers the setting of conformal prediction for uncertainty quantification and studies the relationship between efficiency (size of sets) and entropy (uncertainty of prediction sets). It empirically and theoretically demonstrates that these two objectives are conflicting. The paper proposes a conformal correction approach to achieve improved trade-offs between these two objectives building on the prior work on conformal training. Experiments on image and graph data support the main message of the paper.

The reviewers' appreciated the new findings (identifying this conflict and investigating methods for improved trade-off), but also raised a number of key concerns:
1. Limited scope of theoretical and empirical studies in terms of conformity scoring functions
2. The utility of this trade-off in practical deployment of machine learning classifiers

For 1, the authors' added additional experimental evaluations and acknowledged the limitation of theory for only APS score.
For 2, the response doesn't seem to be satisfactory. Prior work has motivated predictive efficiency for human-AI collaboration where a human goes over all classes in the prediction set to select one. Human studies (https://arxiv.org/abs/2205.01411) have confirmed the utility of efficiency. The example provided in the global response makes assumptions to justify the utility but requires further human studies for validation.

The AC acknowledges the important contribution made by the paper, but the significance/utility concern (#2) is critical. Therefore, I recommend rejecting the paper and strongly encourage the authors to improve the paper for future re-submission.

**Reviewer Concerns:**

1. Limited scope of theoretical and empirical studies in terms of conformity scoring functions
2. The utility of this trade-off in practical deployment of machine learning classifiers

For 1, the authors' added additional experimental evaluations and acknowledged the limitation of theory for only APS score.
For 2, the response doesn't seem to be satisfactory. Prior work has motivated predictive efficiency for human-AI collaboration where a human goes over all classes in the prediction set to select one. Human studies (https://arxiv.org/abs/2205.01411) have confirmed the utility of efficiency. The example provided in the global response makes assumptions to justify the utility but requires further human studies for validation.

**Reviewer Scores:**

Reviewer 3wJp: 4 => 5
Reviewer G9pT: 4 => 5
Reviewer s5ab: 4 => 6
Reviewer wJCq: 4 => 5

---

### Decision · Program_Chairs · 2026-01-26

Reject